# Aging affects the balance of neural entrainment and top-down neural modulation in the listening brain

Molly J. Henry[1,2], Björn Herrmann[1,2], Dunja Kunke[1] & Jonas Obleser[1,3]

Healthy aging is accompanied by listening difficulties, including decreased speech comprehension, that stem from an ill-understood combination of sensory and cognitive changes. Here, we use electroencephalography to demonstrate that auditory neural oscillations of older adults entrain less firmly and less flexibly to speech-paced ($\sim$3 Hz) rhythms than younger adults' during attentive listening. These neural entrainment effects are distinct in magnitude and origin from the neural response to sound *per se*. Non-entrained parieto-occipital alpha (8–12 Hz) oscillations are enhanced in young adults, but suppressed in older participants, during attentive listening. Entrained neural phase and task-induced alpha amplitude exert opposite, complementary effects on listening performance: higher alpha amplitude is associated with reduced entrainment-driven behavioural performance modulation. Thus, alpha amplitude as a task-driven, neuro-modulatory signal can counteract the behavioural corollaries of neural entrainment. Balancing these two neural strategies may present new paths for intervention in age-related listening difficulties.

[1] Max Planck Research Group 'Auditory Cognition', Max Planck Institute for Human Cognitive and Brain Sciences, Stephanstrasse 1a, Leipzig 04103, Germany. [2] Brain and Mind Institute, Department of Psychology, The University of Western Ontario, 1151 Richmond St., London, Ontario, Canada N6A 3K7. [3] Department of Psychology, University of Lübeck, MFC 8, Maria-Goeppert-Strasse 9a, Lübeck 23562, Germany. Correspondence and requests for materials should be addressed to M.J.H. (email: mhenry55@uwo.ca).

Speech comprehension decreases during healthy aging, especially when speech is fast[1,2] or presented against background noise[3–5]. These age-related listening and comprehension difficulties are likely the consequence of an interaction between sensory and cognitive changes[6,7]. For example, perceptually segregating talkers in a 'cocktail-party' situation (that is, attending to one talker against a noisy background that includes irrelevant speech[8]) requires top-down selective attention to the to-be-attended talker[9,10]. Recent work indicates that solving the cocktail-party problem also depends on synchronization of low-frequency neural oscillations (that is, entrainment) to the attended speech[11–13]. Here, we investigated the nature of this interaction by employing a vigilance-style target-detection task, in which near-threshold targets occurred randomly over the time course of rhythmic auditory stimuli[14]. We focused on neural entrainment to the (irrelevant) stimulus rhythm as an index of sensory-driven, bottom-up responses and on alpha-band (8–13 Hz) dynamics as an index of top-down neural modulation. We view our results through the lens of an active sensing framework, which posits a distinction between rhythmic-mode and continuous-mode processing[15,16].

Neural oscillations are entrained by behaviourally relevant rhythmic stimuli. In turn, neuronal excitability is relatively high during temporally predictable, information-carrying parts of the signal and is relatively low during unimportant parts of the signal. As a result, the strength of neural entrainment predicts successful speech comprehension[17,18]. This framework, whereby neural entrainment enhances perception, is referred to as rhythmic-mode processing. Surprisingly, thus far, entrainment in the aging brain has rarely been studied. Aging is accompanied by reductions in neural entrainment of the auditory brainstem response at very fast rates (~100–1,000 Hz; refs 19–23) and a restriction of the range of speech-relevant rates within which older individuals can behaviourally synchronize[24]. Thus, an intriguing possibility is that age-related speech-comprehension deficits might be partially attributable to changes in neural entrainment at speech-relevant rates.

In contrast to sensory-driven neural entrainment, parieto-occipital oscillatory activity in the alpha-frequency band (~8–12 Hz) has been shown to index cognitive processing and effortful listening. Alpha oscillations are modulated by attention deployed in space[25–29], in time[30,31], between sensory modalities[32,33], as well as by task difficulty[30,34,35] and thus are thought to reflect an adaptive, intentional, top-down suppression of task-irrelevant information[26,29,36]. As such, we interpret tonic changes in alpha power as aligning with continuous-mode processing, in which vigilance-style monitoring or attention to a particular time point based on hazard-function style timing are beneficial strategies. Critically, alpha dynamics change substantially with age. Individual alpha frequency slows with increasing age[37], and resting-state alpha oscillations are less likely to spontaneously entrain in older than in younger adults[38]. Older adults show aberrant alpha lateralization during spatial attention tasks[39], in particular for high memory loads[40], and show less alpha modulation than younger adults in response to predictive cues[41,42]. However, in the context of effortful listening to speech, alpha amplitude is modulated more by acoustic degradation for older compared to younger listeners[43] and for individuals with moderate hearing loss compared to normal-hearing listeners[44].

The current electroencephalography (EEG) study examined the dynamic relationship between neural entrainment to stimulus rhythm (during active and passive listening) on the one hand and task-induced alpha modulations on the other, that is, between rhythmic- and continuous-mode processing during vigilance-style monitoring of a rhythmic stimulus. Younger and older participant groups either detected near-threshold gaps in frequency-modulated sounds or listened to those same sounds with instructions to ignore them. A frequency modulation (FM) rate of 2.8 Hz was chosen because this rate is representative of frequency fluctuations in natural speech corresponding to intonation contour[45]. Critically, we also recorded event-related potentials (ERPs) in response to tone sequences that had presentation rates and spectral ranges matched to the frequency-modulated sounds. The reason for this was to rule out the possibility that weaker entrainment for older compared to younger adults could be simply attributed to overall decreases in auditory cortical responsiveness with age.

We hypothesized that older adults would show weaker entrainment to frequency-modulated stimuli than younger adults, and that entrainment might be less strongly modulated by attention for older compared to younger adults. We also expected to see a relationship between entrained neural phase and psychophysical performance, in that psychophysical performance (here, gap-detection) would fluctuate coupled to entrained pre-target neural phase[14]. Since we predicted entrainment strength differences between age groups, we anticipated that older adults would show reduced behavioural modulation by pre-target neural phase, that is, that older adults would be less able to engage in rhythmic-mode processing; however, this prediction was not borne out. With respect to alpha amplitude, we expected age-related differences in top-down auditory attention/inhibition, as indexed by alpha amplitude dynamics. However, given previous mixed results on age-related alpha differences and their relations to behaviour, we did not have strong hypotheses regarding the relationship of alpha amplitude and behavioural modulation, or potential interactions between alpha amplitude and neural entrainment.

The results demonstrate that alpha oscillations effectively shield the listening brain from obligatory behavioural entrainment by the stimulus rhythm: In keeping with a listener's engagement in continuous-mode processing, higher alpha amplitude was associated with a reduced degree of behavioural modulation by entrained neural phase. Moreover, our results reveal that the nature of this tradeoff between neural entrainment to a stimulus rhythm (related to rhythmic-mode processing) and top-down neural modulation (related to continuous-mode processing) changes with age.

## Results

**Neural dynamics during active and passive listening.** Younger and older listeners detected near-threshold gaps embedded in 2.8-Hz frequency-modulated sounds during an active block, and listened to the same sounds without performing a task during a passive block (order of presentation was counterbalanced; Fig. 1a). We assessed low-frequency neural entrainment at 2.8 Hz, and compared the degree to which gap-detection performance was modulated by entrained neural phase (that is, neuro-driven behavioural modulation) between younger and older adults. We also quantified dynamics of alpha-band oscillations during active and passive listening, and correlated alpha amplitude with neuro-driven behavioural modulation. Finally, we recorded evoked responses to a 2.8 Hz tone sequence (Fig. 1a) to compare amplitudes and topographies of entrained versus evoked neural responses.

**Younger and older listeners do not differ behaviourally.** Gap durations were individually adjusted using an adaptive-tracking procedure; thus mean performance was near threshold and did not differ significantly between age groups (hit rates: t(38) = − 1.62, P = 0.11, $r_e$ = 0.25 (independent-samples t-test,

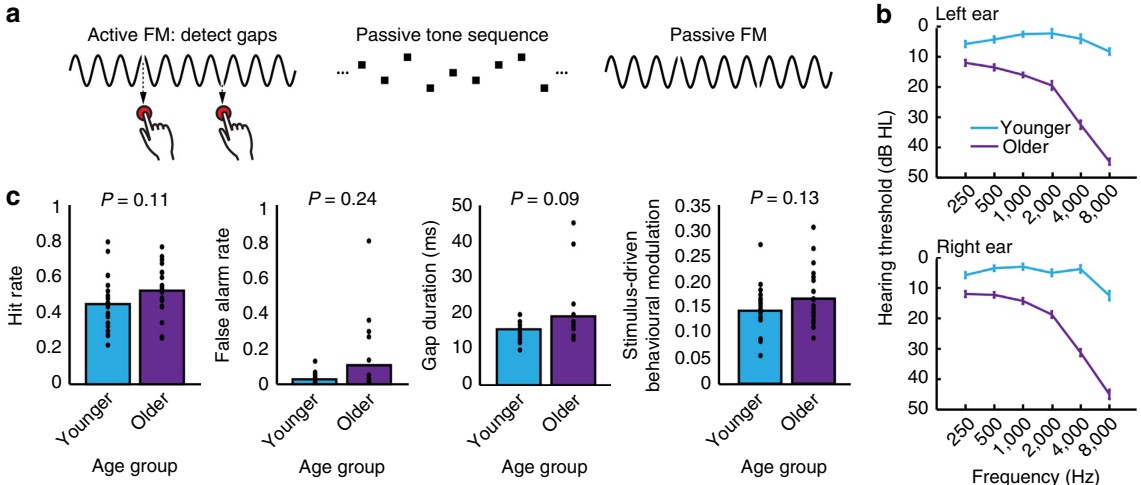

**Figure 1 | Entrainment was tested using frequency-modulated stimuli and tone sequences.** (**a**) Stimuli and experimental design. Participants were exposed to two blocks of FM stimulation; during an active block, they detected near-threshold gaps, while during a passive block they ignored the same stimuli (in counterbalanced order). Between FM blocks, participants passively listened to an 8-min block of tone stimulation (1,400 tones), with presentation rate and frequency range matched to the rate and modulation depth of the FM stimuli. (**b**) Audiograms for younger (blue) and older (purple) listeners shown separately for the left (top) and right (bottom) ears. Both participant groups had average hearing thresholds better than 20 dB up to 2,000 Hz. (**c**) Younger and older listeners did not differ significantly according to any behavioural measure.

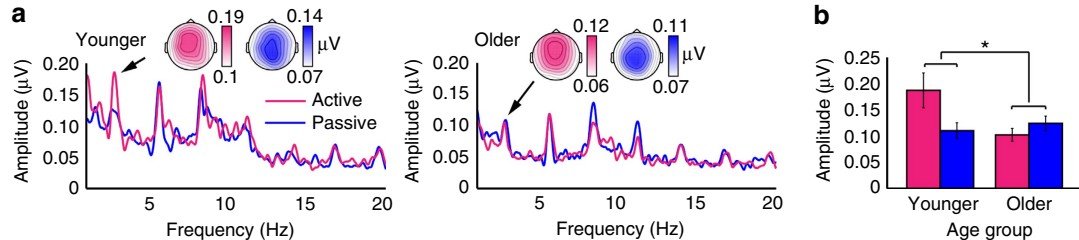

**Figure 2 | Attention effects on entrainment were larger for younger than for older adults.** (**a**) Frequency-domain representations of the EEG signal for active (pink) and passive (blue) listening to FM stimuli, shown separately for younger (left) and older (right) averaged over a fronto-central electrode cluster: Fz ,F3, F4, FC3, FC4, Cz, C3, C4. (**b**) 2.8 Hz spectral amplitude: a significant interaction between age and attention resulted from stronger effects of attention (active minus passive) for younger as compared to older adults ($P = 0.04$). Error bars show s.e.m.

$n = 20$ younger, $n = 20$ older); false alarm rates: $z = 1.18$, $P = 0.24$, $r = 0.19$ (rank-sum test, $n = 20$ younger, $n = 20$ older); Fig. 1c). Equalizing performance across age groups required presenting older listeners with slightly (but not significantly) longer gaps than younger listeners ($z = 1.35$, $P = 0.09$, $r = 0.27$ (rank-sum test, $n = 20$ younger, $n = 20$ older); Fig. 1c), in line with previous research indicating that gap-detection thresholds become worse with age[46,47]. The degree to which gap-detection performance depended on the phase of the FM into which the gap fell (stimulus-driven behavioural modulation) did not differ between age groups ($t(38) = -1.54$, $P = 0.13$, $r = 0.24$ (independent-samples $t$-test, $n = 20$ younger, $n = 20$ older); Fig. 1c).

**Sensory-driven entrainment differs between age groups.** Sensory-driven entrained neural responses were calculated using a fast Fourier transform (FFT). Spectral amplitudes (µV) from the 2.8-Hz frequency bin (Fig. 2a) were submitted to a 2 × 2 mixed-measures analysis of variance (ANOVA) with the factors age (between-participants, younger ($n = 20$), older ($n = 20$)) and attention (within-participant, active, passive), which yielded a main effect of age ($F(1,38) = 4.42$, $P = 0.04$, $\eta_p^2 = 0.10$) and an Age × Attention interaction ($F(1,38) = 4.60$, $P = 0.04$, $\eta_p^2 = 0.11$); the main effect of attention did not reach significance ($F(1,38) = 2.74$, $P = 0.11$, $\eta_p^2 = 0.07$). The interaction resulted

from the effect of attention (that is, the difference between active and passive conditions) being larger for younger than for older adults (Fig. 2b; $t(38) = 2.14$, $P = 0.04$, $r = 0.33$ (independent-samples $t$-test, $n = 20$ younger, $n = 20$ older)), in line with older adults lacking entrainment flexibility. Moreover, younger adults' entrainment strength was significantly greater than older adults' for the active ($t(38) = 2.42$, $P = 0.02$, $r = 0.37$ (independent-samples $t$-test, $n = 20$ younger, $n = 20$ older)), but not for the passive condition ($t(38) = 0.69$, $P = 0.49$, r = 0.11 (independent-samples $t$-test, $n = 20$ younger, $n = 20$ older)). A parallel ANOVA on spectral amplitudes from the 5.6 Hz (that is, second harmonic) bin did not yield any modulation by either age or attention (all $P \geqslant 0.19$).

**Age effects are not due to overall cortical responsiveness.** ERPs evoked by individual tones in sequences that had presentation rates and spectral ranges matched to the frequency-modulated sounds were examined to rule out the possibility that entrainment differences between younger and older adults could be due to a generalized reduction in cortical responsiveness to sound in older adults[22,48]. Separate comparisons between age groups for ERP amplitudes in the range of the P1, N1 and P2 components (independent-samples $t$-tests, $n = 20$ younger, $n = 20$ older) revealed significant differences for the N1 and P2 (P1: $t(38) = -1.63$, $P = 0.11$,

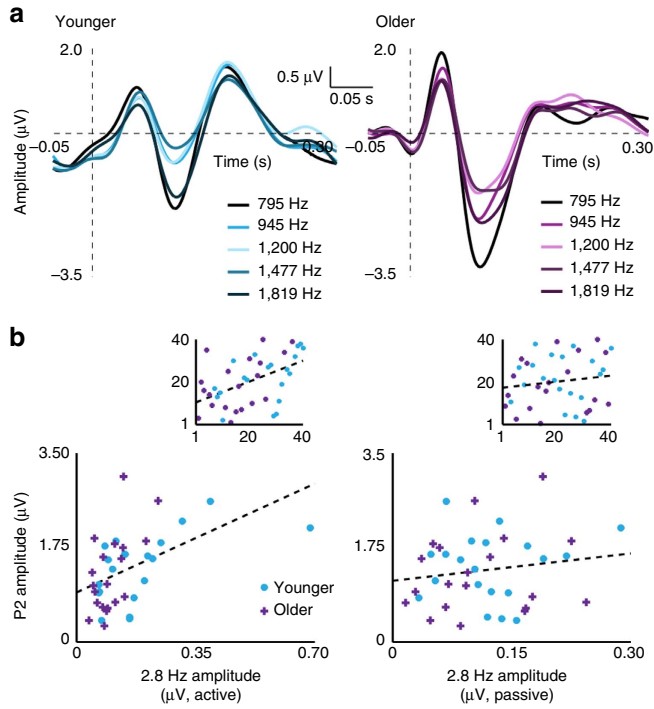

**Figure 3 | ERPs in response to individual tones. (a)** Older adults showed enhanced N1 responses and reduced P2 responses relative to younger adults. Individual lines show ERPs to each tone frequency. **(b)** P2 amplitude was significantly correlated with 2.8 Hz amplitude measured during active task performance (left), but not with 2.8 Hz amplitude measured during passive listening (right). Insets show plots of ranks on which Spearman correlations were based. Ranks on 2.8 Hz amplitude are plotted on the *x* axis and ranks on P2 amplitude on the *y* axis.

$r = 0.26$; N1: $t(38) = 3.21$, $P = 0.003$, $r = 0.46$; P2: $t(38) = 2.25$, $P = 0.03$, $r = 0.34$; Fig. 3a). In particular, the N1 component was larger for older adults while the P2 component was smaller.

Since the direction of the age effect on the P2 was the same as the age effect on entrainment strength, we correlated P2 amplitudes with active and passive entrained 2.8 Hz amplitudes (we did the same for P1 and N1 amplitudes and found no significant relationships, all $p_{FDR} \geq 0.31$ (Spearman rank-order correlations, $n = 40$)). P2 amplitudes were significantly correlated with 2.8 Hz amplitude in the active ($\rho = 0.49$, $p_{FDR} = 0.005$ (Spearman rank-order correlation, $n = 40$); Fig. 3b), but not the passive condition ($\rho = 0.14$, $p_{FDR} = 0.58$ (Spearman rank-order correlation, $n = 40$); Fig. 3b). Thus, although N1 amplitudes were actually larger for older compared to younger adults[48,49], P2 amplitudes were reduced in older adults and correlated with the magnitude of 2.8 Hz entrainment. This held also for separate analyses in younger ($\rho = 0.54$, $P = 0.02$ (Spearman rank-order correlations, $n = 20$)) and in older adults ($\rho = 0.43$, $P = 0.06$ (Spearman rank-order correlations, $n = 20$)), ruling out the possibility that the overall correlation reflects age effects on both variables.

Interestingly though, this correlation was present only for active entrainment, and not for the passive condition, despite P2s being measured under passive listening conditions as well. In an effort to better characterize whether entrained neural responses and P2s may have arisen from the same neural generators, we performed an in-depth analysis of topographies of different neural response types.

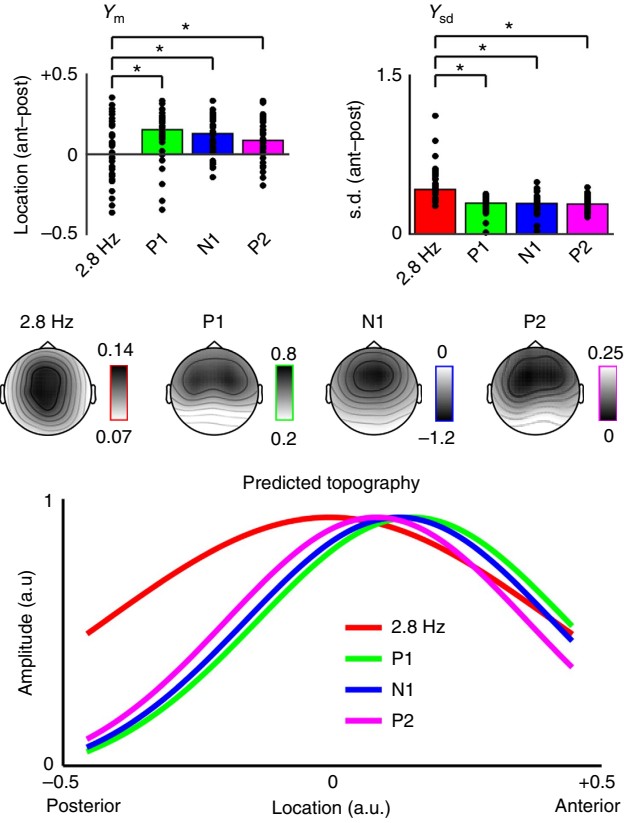

**Figure 4 | Topographies differ for entrained neural responses (2.8 Hz) and ERPs.** Actual topographies for each neural response type are shown in the centre of the plot (arranged horizontally, 2.8 Hz, P1, N1 and P2, responses; note that for this analysis the N1 topography was sign-inverted). Bar graphs (top) summarize the median (bar) and single-participant (black dots) data for each type of neural response and each fitted Gaussian parameter ($y_m$ on left and $y_{sd}$ on right). Note: two outliers are not shown for the 2.8 Hz condition for parameter $y_{sd}$ (values were 8.15 and 40.24). Line plot (bottom) shows predicted topographies for each neural response type in the anterior-posterior plane (right) based on median parameter values over participants. Entrained-response topographies were dissociable from ERP-component topographies.

**Entrainment generators are discriminable from auditory ERPs.** We also compared topographies of the entrained 2.8 Hz neural response to ERP-component topographies for the P1, N1 and P2 (Fig. 4). The analysis of topographical distributions allowed us to test whether the generators of neural responses that were entrained at 2.8 Hz would be dissociable from the generators of ERPs evoked by tones presented at 2.8 Hz. This analysis speaks to the ongoing debate in the literature as to whether peaks in EEG spectra in response to modulation reflect entrainment of ongoing (spontaneous) neural oscillations[50,51] or whether the same spectral peaks might instead reflect the summation of a series of invariant transient neural responses evoked by rhythmic stimulation independent of ongoing oscillatory activity[52].

First, we tested whether age affected topographies for any neural response type. Anterior-posterior topography centres and their standard deviations ($y_m$ and $y_{sd}$, respectively) were quantified via two-dimensional Gaussian fits and then submitted to separate nonparametric rank-sum tests (($n = 20$ younger, $n = 20$ older) false-discovery rate (FDR) -corrected across neural response types). None of the tests revealed an age effect (all $p_{FDR} \geq 0.09$, all $r < 0.27$). Next, we confirmed that attention did not significantly affect the topography for the 2.8 Hz

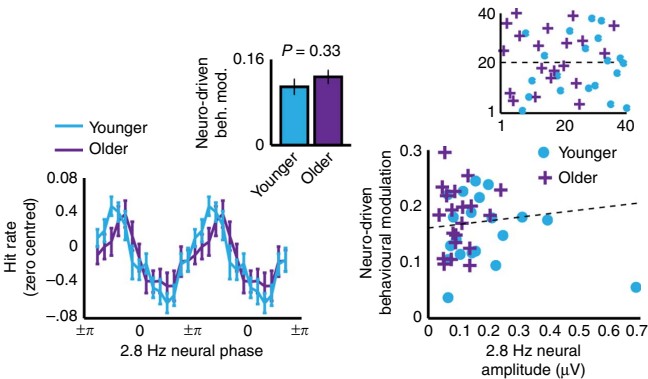

**Figure 5 | Gap-detection is modulated by pre-gap neural phase.** Left: Neuro-driven behavioural modulation: hit rates (shown zero-centred) were modulated by 2.8 Hz neural phase with a similar magnitude (neuro-driven behavioural modulation, that is, hit rate peak-trough) for younger and older participants (inset). Error bars show s.e.m. Right: Neuro-driven behavioural modulation was not correlated with entrainment strength. Best-fit line shown ignoring outlying data point (from the younger group). Inset plots ranks on which Spearman correlation was based. Ranks on 2.8 Hz amplitude is plotted on $x$ axis and ranks on neuro-driven behavioural modulation are plotted on $y$ axis.

entrained neural responses (active versus passive) using separate Wilcoxon sign-rank tests for each model parameter (FDR-corrected across parameters), all $p_{FDR} \geqslant 0.25$, all $r < 0.18$ ($n = 40$). For subsequent statistical comparisons of neural responses, we collapsed across age and attention conditions.

Separate Wilcoxon sign-rank tests for 2.8 Hz topography versus each ERP-component topography were conducted for the fitted Gaussian parameters. The entrained 2.8 Hz topography was dissociable from ERP topographies based on both Gaussian parameters describing topographies in the anterior-posterior plane. The entrainment topography differed from all ERP-component topographies in terms of both centre, $y_m$, (all $p_{FDR} \leqslant 0.04$, all $r \geqslant 0.33$) and standard deviation, $y_{sd}$, (all $p_{FDR} \leqslant 0.0001$, all $r \geqslant 0.58$). Thus the entrainment topography was centred less anterior and spread wider in the anterior-posterior direction than any of the ERP topographies. Dissociation of the topographical distributions of the two response types provides evidence against the assertion that spectral peaks would necessarily reflect only a series of evoked responses.

**Entrainment's behavioural influence is similar across age.** Neuro-driven behavioural modulation (that is, modulation of behaviour by 2.8 Hz neural phase) did not differ significantly between age groups ($z = -1.31$, $P = 0.19$, $r = 0.21$ (rank-sum test, $n = 20$ younger, $n = 20$ older); Fig. 5). An assumption we held (and one held at least implicitly by many other studies on neural entrainment) is that stronger entrainment by the 2.8 Hz FM would lead to stronger neuro-driven behavioural modulation. However, neuro-driven behavioural modulation was not directly correlated with 2.8 Hz spectral amplitude ($\rho = 0.002$, $P = 0.50$). The section 'Entrainment and neural modulation affect behaviour oppositely' will elaborate on this apparent null finding.

**Dynamics of top-down neural modulation differ with age.** Younger and older adults presented with qualitative differences in alpha dynamics (Fig. 6a). Indeed, the age × attention interaction was statistically significant across almost the entire time course (all $p_{FDR} < 0.0446$ all $\eta_P^2 > 0.11$ ($2 \times 2$ mixed-measures ANOVA performed at each time point, $n = 20$ younger, $n = 20$ older)) and

remarkably stable across trial time. For both younger and older adults, alpha amplitude stayed near baseline values during passive listening. For younger adults, alpha amplitude increased significantly during active task performance compared to passive listening to the same stimuli over the entire time course of stimulation (all $p_{FDR} < 0.0497$, all $r > 0.44$ (paired-samples $t$-test performed at each time point, $n = 20$)). In contrast, for older adults, alpha amplitude started to decrease during active task performance relative to passive listening $\sim 3$ s into the stimulus (all $p_{FDR} < 0.0493$, all $r > 0.48$ (paired-samples $t$-test performed at each time point, $n = 20$)).

Since we observed age differences in both entrainment strength and alpha amplitude, we correlated neuro-driven behavioural modulation with both neural measures. First, the direct correlation between alpha amplitude and entrainment strength was significant ($\rho = 0.54$, $P < 0.001$ (Spearman rank-order correlation, $n = 40$); Fig. 6b). Interestingly, however, neuro-driven behavioural modulation was also significantly (negatively) correlated with tonic alpha power averaged over the 10 s time course ($\rho = 0.45$, $P = 0.002$ (Spearman rank-order correlation, $n = 40$)). That is, individuals with higher alpha amplitude showed weaker behavioural modulation by entrained neural phase.

**Entrainment and neural modulation affect behaviour oppositely.** We failed to observe the predicted relationship between entrainment strength (2.8 Hz amplitude) and neuro-driven behavioural modulation when we did not consider alpha amplitude. However, we observed that higher alpha amplitude was associated with a decrease in the degree to which behaviour was modulated by entrained neural phase (Fig. 6b). Thus, we calculated a partial correlation between entrainment strength and neuro-driven behavioural modulation, partialling out alpha amplitude. This correlation was statistically significant ($\rho_P = 0.30$, $P = 0.03$ (partial Spearman rank-order correlation, $n = 40$); Fig. 7).

**Discussion**
The current work demonstrates that neural entrainment to environmental rhythms is weaker in older compared to younger adults, while they perform a difficult gap-detection task. In particular, younger adults' entrainment increased with attention to the sounds (compared to passive listening) to a greater extent than did entrainment strength in older adults. Simultaneously, while actively performing this difficult gap-detection task (relative to passive listening), the amplitude of alpha oscillatory activity was enhanced for younger adults but suppressed for older adults.

Across groups, we observed complementary roles of entrained and alpha-band neural oscillations on performance. Stronger entrainment led to greater modulation of behavioural performance by neural phase, while alpha amplitude counteracted this relationship. The rhythmic influence of neural entrainment on detection behaviour was thus reduced for individuals with higher alpha amplitude. These data thus suggest that reliance on a rhythm to guide attention (via neural entrainment) versus engaging endogenous attentional mechanisms (as indexed by alpha amplitude) reflect two distinct neural strategies for the same difficult task.

We note that entrainment strength and alpha amplitude were positively correlated across participants; individuals with stronger entrainment also had higher alpha amplitudes. This finding may seem incongruent with a proposed tradeoff between the two neural strategies[53]. However, it is important that although increased alpha amplitude does not reduce entrainment strength, it does reduce entrainment's effects on behaviour. That is, predicting how strongly a listener's behaviour depends on

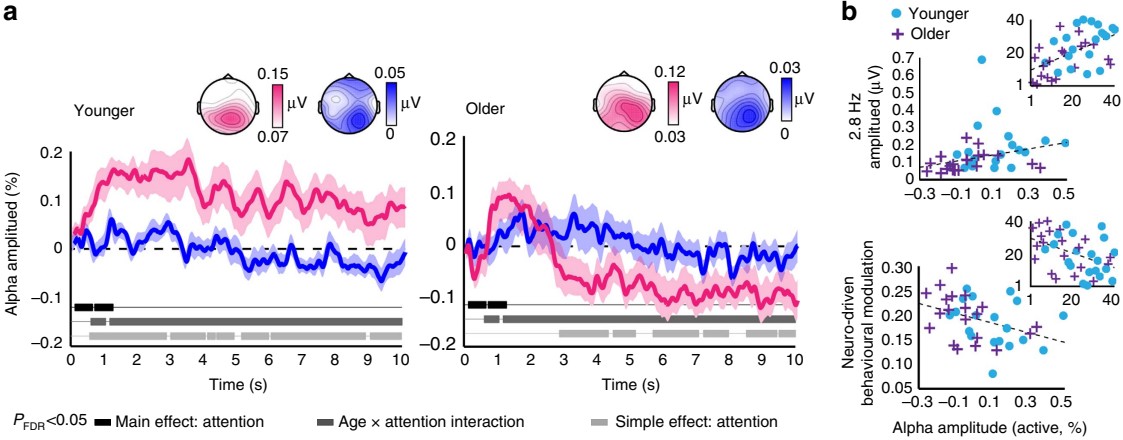

**Figure 6 | Alpha amplitude dynamics differed between younger and older adults.** (**a**) Younger adults showed a significant increase in alpha amplitude during active task performance compared to passive listening (left), while older adults (right) showed the opposite effect. FDR-corrected significance is shown for the main effects of attention (black) and age × attention interaction (dark grey) on plots of both the younger and older alpha time courses. Simple effects of attention within each age group (light grey) are shown separately for younger and older data. Shaded regions show standard error of the mean. (**b**) Alpha amplitude averaged over the 10-s stimulus time course from electrode Pz correlated positively with 2.8 Hz amplitude averaged over a fronto-central electrode cluster (Fz ,F3, F4, FC3, FC4, Cz, C3, C4; top) and negatively with neuro-driven behavioural modulation (bottom). Insets show ranks on which Spearman correlations were based. Ranks on alpha amplitude are plotted on the x axis. Y axis shows ranks on 2.8 Hz amplitude (top) and on neuro-driven behavioural modulation (bottom).

entrained neural oscillations requires knowing about both the strength of entrainment as well as how effectively alpha oscillations will shield behaviour from entrainment.

Older adults showed weaker entrainment to slow frequency-modulated sounds during a difficult gap-detection task, indicating that indeed neural entrainment at speech-relevant rates worsens with age. Our observation is consistent with previous work demonstrating that, with age, both cortical envelope-following responses to amplitude-modulated stimuli[54] and auditory brain stem responses are less consistently phase-locked to the temporal stimulus structure[19,20,22]. In contrast, previous work examining neural entrainment to FM demonstrated that, although younger and older adults had similar response magnitudes at 4 Hz, older adults' entrained neural responses exceeded those of younger adults at slightly higher rates (16–32 Hz)[55]. Moreover, studies using relatively slow, discrete auditory stimulation (for example, tone sequences, an extreme form of amplitude modulation, AM) have shown that age leads to increased neural response magnitudes and phase-locking precision[22,48]. A similar age-related increase in neural responses to amplitude-modulated stimuli was recently reported for a 4 Hz modulation rate and 100% modulation depth[56]. Moreover, recent work using an approach that involved reconstructing speech envelopes from brain responses showed exaggerated cortical representations of speech compared to younger adults, despite depressed brainstem responses[57,58].

What can be the source of these apparent inconsistent age differences in the literature? Several factors have been shown to modulate age effects on neural entrainment, including the presence[54] and content[57] of background noise as well as the rate[23,47,56,59] and depth[55] of the stimulus modulation. Previous work on neural entrainment to AM or FM has used mostly passive listening paradigms, and non-human studies often test animals under anaesthesia. Here, we did not find a difference between younger and older adults' brain responses during passive listening to FM[55], but only when attention was directed towards a demanding auditory task. Thus, attention is another factor that should be considered when comparing auditory cortical responses across age groups and which is relevant for generalization to everyday listening situations.

It is also important to consider whether a given study operationalized temporal modulation for neural entrainment via amplitude or frequency fluctuations. Entrained neural responses to AM and FM are dissociable in humans[60,61] and non-human animals[62]. Previous work from our laboratory moreover indicates that, while the time course of cortical adaptation to repeated sounds contracts with age[49], the pattern of cortical frequency-specific adaptation does not differ between younger and older adults[48]. Lastly, comparisons of topographies for entrained responses to FM versus evoked responses to tone-sequence stimulation suggest that the two response types originate from non-redundant cortical generators. Taken together, these results point to differences in the cortical representations of sounds in which temporal structure is communicated by AM versus FM, and age-related changes thereof. We suggest that this dissociation at the cortical level would translate to age-related differences in the relative usefulness of syllabic versus prosodic information for speech comprehension, which may ultimately be exploitable by future hearing-aid algorithms.

Our tone-stimulation block revealed larger N1 ERP components for older compared to younger adults[22,49]. Thus, weaker and less flexible entrainment to FM with age happens alongside stronger N1 responses to tones. However, we also observed smaller P2 amplitudes for older adults, and this P2 amplitude was correlated with 2.8 Hz amplitude as measured during active performance of the gap-detection task. Interestingly, however, P2 amplitudes were not correlated with passive 2.8 Hz amplitude, despite ERPs to tones being measured during passive listening. Thus, these data render it unlikely that weaker entrainment for older compared to younger adults is attributable to overall decreases in cortical responsiveness with age.

Moreover, the analysis of topographical distributions provides additional evidence that the generators of 2.8 Hz entrained neural generators are not identical to the generators of ERPs evoked by tones presented at 2.8 Hz. These results speak to the ongoing debate in the literature as to whether peaks in EEG spectra in response to modulation reflect entrainment of ongoing (spontaneous) neural oscillations[50,51] or whether the same spectral peaks might instead reflect the summation of a series

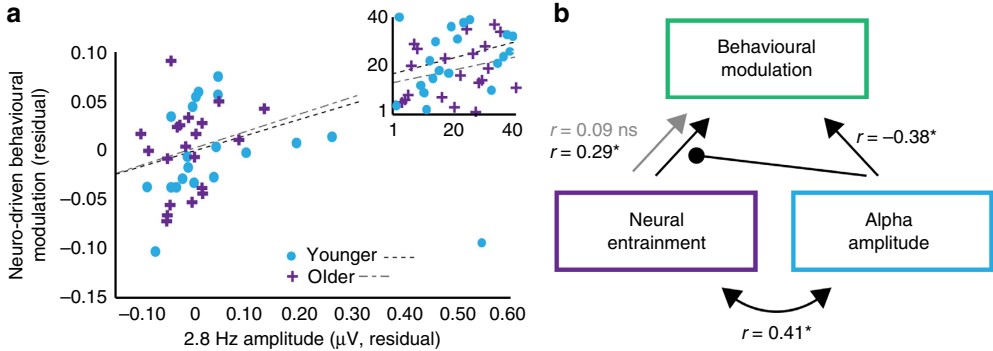

**Figure 7 | Entrainment strength predicts neuro-driven behavioural modulation.** (**a**) The partial correlation after partialling out alpha amplitude from the Pz electrode between 2.8 Hz amplitude (averaged over a fronto-central electrode cluster: Fz, F3, F4, FC3, FC4, Cz, C3, C4) and neuro-driven behavioural modulation was significant. Plot shows residuals, that is, deviations of the neuro-driven behavioural modulation observations from the fitted function characterizing the relationship between alpha amplitude and neuro-driven behavioural modulation, plotted against the residuals from the correlation between alpha amplitude and 2.8 Hz amplitude. Best-fit lines are shown separately for younger and older groups. Line for younger group excludes the extreme data point marked with arrow. (**b**) Schematic representation illustrating the statistical interaction between neural entrainment, alpha amplitude and behavioural modulation. Behavioural modulation was significantly predicted by neural entrainment strength, but only when alpha amplitude was partialled out. Inset shows ranks on which Spearman correlation was based. *X* axis shows ranks on 2.8 Hz amplitude residuals, and *y* axis shows ranks on neuro-driven behavioural modulation residuals.

of invariant transient neural responses evoked by rhythmic stimulation independent of ongoing oscillatory activity[52]. Dissociation of the topographical distributions of the two response types provides evidence against the assertion that spectral peaks will necessarily reflect a series of evoked responses.

Alpha amplitude dynamics differed during active performance of the gap-detection task compared to passive listening for both age groups. However, these changes were in opposite directions for younger and older adults—younger adults' alpha amplitude during task performance increased relative to passive listening, whereas older adults' alpha amplitude was significantly suppressed. Previous work on younger adults suggests that increases in parieto-occipital alpha amplitude reflect engagement of top-down inhibitory processes that allow for suppression of task-irrelevant information[36]. Here, we suggest that enhanced alpha amplitude during task performance reflects selective inhibition of the rhythmic information conveyed by the FM, in an effort to keep performance consistently high over time (referred to above as a shielding role of alpha). This hypothesis is supported by the negative correlation between alpha amplitude and neuro-driven behavioural modulation—the higher a participant's alpha amplitude, the weaker the influence of rhythm on gap-detection performance.

A number of previous studies have demonstrated differences in alpha dynamics between younger and older listeners. For example, younger adults show increased pre-stimulus alpha in anticipation of the need to suppress irrelevant information, whereas older adults do not[39,42]. This is despite older adults showing normal behavioural performance[42] and normal ERP-based indices of attentional orienting[39]. Moreover, older adults have smaller alpha modulation in response to predictive cues regarding the timing of upcoming stimulus events[41] alongside smaller behavioural benefits of temporal cues than younger adults[41,63]. Such results have led to the hypothesis that older adults potentially use a different strategy than younger adults to filter out irrelevant sensory information, one not based on alpha oscillations[42]. This suggestion is particularly interesting in light of evidence that different strategies for shifting and maintaining attention can be observed even in young participants—in particular, whether or not an individual's alpha power was modulated in anticipation of a visual target was predictable from their resting alpha-power level[64]. Although we

did not test for age differences in resting alpha power in the current study, it is possible that different strategies used by younger and older listeners may have been in part predictable from resting alpha levels. However, it is not obvious that age necessarily diminishes alpha modulation *per se*, as older adults' alpha amplitude is more strongly modulated than younger adults' by increasing acoustic degradation[43]. One factor that seems to mediate age differences in alpha dynamics is difficulty: older adults' alpha dynamics differ more markedly from younger adults' when task complexity[41] and difficulty[40] are high.

In the current study, the time courses of younger and older adults' alpha amplitude modulation differed. Specifically, both groups showed increased alpha amplitude during active task performance (compared to passive listening) at the beginning of the auditory stimulus. Approximately 3 s into the stimulus, older adults' alpha amplitude became significantly suppressed. We would like to suggest that, given the difficult nature of the gap-detection task, three seconds reflects a kind of breaking point, and that older adults were unable to maintain an alpha-based shielding strategy to ignore the stimulus rhythm. We hypothesize that the switch to alpha suppression reflects the switch to a qualitatively different strategy for task performance, here allowing attention to fluctuate rhythmically (that is, a switch to rhythmic-mode processing)[15]. This might explain why, despite older adults entraining less strongly to the auditory rhythm than younger adults, we observed no age differences in neuro-driven behavioural modulation. Thus, we propose that stimulus-driven rhythmic attention (entrainment) and top-down alpha-driven attention play complementary roles in auditory perception, echoing the previously proposed distinction between rhythmic-mode and continuous-mode processing[15].

The current study aimed to test the hypothesis that sensory-driven neural entrainment, as well as top-down neural modulation, changes with age. We reasoned that age-related changes in either measure might contribute to listening and speech comprehension difficulties, and in particular difficulty solving the cocktail-party problem. However, the current study neither makes use of naturalistic speech stimuli[57,58], nor does it attempt to recreate the cocktail-party problem by including distractor sounds. Thus, one important future direction is to extend this work using stimuli and paradigms that better approximate the naturalistic sounds and situations that cause

problems for older adults (for example, competing speech, background noise, time-compressed speech). Using such manipulations to create a situation in which older adults cannot default to a rhythmic-mode strategy when a top-down, alpha-based strategy fails might clarify the contributions to age-related speech comprehension deficits.

The timescale of the effects discussed in the current study is relatively coarse; we have aggregated over all trials to observe a tradeoff between the effects of entrainment and alpha amplitude on behaviour. However, a more fine-grained examination of trial-by-trial tradeoffs between measures could reveal more subtle dynamical shifts between strategies or processing modes[53]. Such an analysis would necessitate a behavioural measure other than neuro-driven behavioural modulation, as this particular measure requires aggregation over many trials to calculate.

Finally, an important future direction is to examine the impact of hearing loss (more severe than in our older sample) on neural entrainment to speech rhythm and the relationship of entrainment to speech comprehension. Hearing loss changes the frequency content arriving at the cortex, and manipulations such as noise vocoding that degrade acoustics result in reductions in entrainment strength[17,18]. Moreover, a leading theory of hearing loss and cognition suggests that cognitive deficits can stem from hearing loss[6,7]; thus it is possible that the balance between sensory-driven neural entrainment and top-down neural modulation is further shifted with moderate-to-severe hearing loss[65].

To conclude, the current study demonstrates that neural oscillations in older adults are entrained less strongly and less flexibly than in younger adults by rhythmic stimuli with rates in speech-relevant ranges. Alpha oscillatory dynamics also differ between younger and older listeners—alpha increased for younger listeners but decreased for older listeners during active task performance (as compared with passive listening to the same stimuli). These findings give credence to the idea that some of older adults' difficulty listening and understanding speech under adverse listening conditions stems from a combination of age-related changes to the fidelity or flexibility of sensory-driven neural entrainment and changes to the ability to inhibit irrelevant auditory information—two processes whose interaction is here shown to determine auditory perception.

## Methods

**Participants.** Forty individuals (20 younger (10 male and 10 female), age 18–31 years, M = 25.4 years ± s.d. = 3.3 years; 20 older (8 male and 12 female), age 61–77 years, M = 67.3 years ± s.d. = 5.3 years) took part in the experiment. All participants gave written informed consent and received financial compensation. The procedure was approved of by the ethics committee of the medical faculty of the University of Leipzig and in accordance with the declaration of Helsinki.

**Stimuli.** Auditory stimuli were generated by MATLAB software (Mathworks, Inc.) at a sampling rate of 60,000 Hz. In close analogy to stimuli used previously[14], frequency-modulated stimuli (FM stimuli) were 10 s narrow-band noises modulated at a rate of 2.8 Hz and a depth of 33.5% (Fig. 1a). The centre frequency of the complex carrier signal was randomized from trial to trial and took on values ranging uniformly between 1,000 and 1,400 Hz. All stimuli comprised 30 frequency components sampled from a uniform distribution with a 500 Hz range around the centre frequency. The amplitude of each component decreased linearly with increasing distance to the centre frequency, according to the following function:

$$A_i = 1 - \left| \frac{cf_i - f_0}{f_0} \right|,$$

where $i$ indexes the individual component, $cf_i$ refers to the individual component frequency and $f_0$ corresponds to the centre frequency. All stimuli were normalized with respect to peak amplitude, and were presented at 50 dB sensation level (SL; hearing thresholds were determined prior to EEG recordings for each participant individually for a 1,200 Hz sine tone using the method of limits and all stimuli were presented 50 dB above the individual hearing threshold to ensure audibility). Two, three, or four near-threshold gaps were inserted into each 10 s stimulus (gap onset and offset were gated with 3 ms half-cosine ramps) without changing the duration of the stimulus. Gaps never occurred in the first or final 1.5 s of the 10 s stimulus,

and were constrained to fall no closer to each other than 1.5 s. Gaps occurred equally often in each of 20 phase bins equally spaced around the FM cycle; a total of 240 gaps was presented.

Tone sequences were composed of 100-ms sine tones taking on 5 unique frequencies varying logarithmically around 1,200 Hz (792, 945, 1,200, 1,477 and 1,819 Hz; centre frequency matched to the mean carrier used for FM stimuli). Tones were presented at a rate of 2.8 Hz in trains of 1,400 (lasting ∼8 min) and the tone frequency varied randomly from tone to tone. Thus, the presentation rate and frequency range were matched as closely as possible between the FM sounds and tone sequences.

**Procedure.** Pure-tone audiograms were measured for all participants at octave frequencies ranging from 250 to 8,000 Hz separately for the left and right ears (Fig. 1b). Participants' gap-detection thresholds were then estimated according to a custom adaptive-tracking procedure so that gap-detection hit rates approximated 50%. After preparation of the EEG, participants heard two blocks of FM stimuli containing short gaps. During an active block, listeners responded each time they detected a gap by pressing a button on a response box (a response was considered a hit when it occurred within 1.5 s after gap onset). During a passive block, participants were instructed to ignore the auditory stimuli that were identical to those presented during the active block including presence and location of gaps. (We note that explicitly instructing participants to ignore the auditory stimuli potentially departs from a classic definition of so-called passive listening. However, many studies measuring neural responses to frequency and amplitude modulation during passive listening allow participants to watch a silent movie, sometimes with subtitles[49,66,67], so it is likely that these situations also involve active ignoring.) The order of active and passive blocks was counterbalanced across participants. Between blocks of FM stimuli, participants listened passively to the 8 min block of tone stimulation. Each participant heard 84 FM stimuli during each active and each passive block (for a total of 168 FM stimuli) and 1,400 tone stimuli. Including adaptive estimation of gap-detection thresholds and preparation of the EEG, the entire experiment lasted ∼3 h.

**Behavioural data acquisition and analysis.** For the active FM block, we calculated a number of dependent measures to confirm that our adaptive-tracking procedure matched the performance of the younger and older participant groups. Hit rates were calculated as the proportion of gaps that were followed by a button press after no more than 1.5 s. False alarm rates were calculated as prescribed by Bendixen et al.[68]. We also retained the gap durations that were used for each participant to approximate 50% detection rates. Hit rate was tested between younger and older participants using an independent-samples $t$-test; false alarm rates and gap durations were compared between groups using separate nonparametric rank-sum tests (both dependent measures for both age groups were significantly non-normal according to Shapiro–Wilk tests). Tests for hit and false alarm rates were two tailed, but the test on gap durations was one tailed based on previous literature indicating that older listeners have higher gap-detection thresholds than younger listeners[46,47]. The degree to which hit rates were modulated by FM phase (referred to as 'stimulus-driven behavioural modulation') was calculated as the maximum minus the minimum hit rate over the 20 FM phase bins after data were smoothed with a five-bin unweighted kernel. It is important to smooth these data because an individual could have a very large maximum-minimum value simply due to noisy data; however, we were interested in systematic relationships between stimulus phase and hit rate. Smoothing penalizes extreme values that are unrelated to neighbouring data points more than those that continue a trend from their neighbours. We tested whether stimulus-driven behavioural modulation differed significantly between younger and older adults using an independent-samples $t$-test.

**EEG data acquisition and analysis.** The EEG was recorded from 26 Ag-AgCl scalp electrodes mounted on a custom-made cap (Electro-Cap International), according to the modified 10–20 system, and additionally from the left and right mastoids. Signals were recorded continuously with a passband of DC to 135 Hz and digitized at a sampling rate of 500 Hz (TMS international, Enschede, The Netherlands). Online reference was the nose and the ground electrode was placed at the sternum. Electrode resistance was kept under 5 kΩ. All EEG data were analysed offline using custom Matlab scripts and Fieldtrip software[69].

Data were preprocessed according to two pipelines. One pipeline was geared towards frequency-domain analysis of full-stimulus epochs (FM stimuli) and analysis of tone-evoked responses (tone-sequence stimuli), and the second pipeline was geared towards analysis of pre-stimulus neural phase in short epochs centred on gap onset (FM stimuli only).

The first pipeline involved high-pass filtering the continuous raw data from each block (active FM, passive FM, passive tone sequence; 0.6 Hz, 1,395 points, Hann window). Data were then divided into individual trial epochs, which for the FM blocks ranged from −1.5 to 11.5 s with respect to FM-stimulus onset and for the tone-sequence block ranged from −1.6 to 1.9 s with respect to each tone onset. Individual trials were then low-pass filtered at 80 Hz (51 points, Hann window) and re-referenced to average mastoids. Data from all three blocks were then combined for independent component analysis (ICA). Blinks, muscle activity,

electrical heart activity, and noisy electrodes were removed from the signal using the Fieldtrip-implemented runica method[70], which performs ICA decomposition using the logistic infomax algorithm[71] with principle component dimension reduction. Individual trials were subsequently removed if the amplitude range exceeded 120 μV; of the 1,568 defined trials (84 for each FM block and 1,400 for the tone-sequence block), the median number of rejected trials was 5 ( ± 10 semi-interquartile range), and did not differ between age groups ($p \approx 1$).

The second pipeline, geared towards the analysis of pre-stimulus 2.8 Hz phase, critically omitted high-pass filtering, and thus first involved epoching the continuous raw data from the active and passive FM blocks ( − 1.5 to 15.5 s) and then low-pass filtering (51 points, Hann window), re-referencing, and removal of the same components as the previously described pipeline using ICA. (We have previously demonstrated that the ICA routine implemented here does not bias estimates of neural phase[72].) We subsequently rejected the same set of trials as was identified by the artefact rejection routine from the previously described pipeline. Following artefact rejection, shorter epochs were defined that ranged between − 1.95 s and + 1.95 s with respect to each gap onset. Further analysis of pre-stimulus phase effects is described below.

For frequency-domain analysis of 2.8 Hz, we considered only the time window ranging from 1 to 9 s relative to stimulus onset in order to avoid contamination by onset- and offset-evoked responses. In order to compensate for the relatively small number of trials in each of the FM blocks, we employed a data resampling procedure that involved randomly selecting 1,000 5.375-s long data segments (corresponding 15 cycles of the 2.8 Hz FM) starting at random time points from the 84 trials in each condition. The 1,000 data segments were then shifted in the time domain, so that the 2.8 Hz FM-stimulus phase would have been aligned across segments. Segments were averaged in the time domain before transformation to the frequency domain (thus, we were examining so-called 'evoked' spectral amplitude). Time-domain data were multiplied with a Hann window and then submitted to a FFT. Subsequently, frequency-domain data were averaged over a fronto-central electrode cluster comprising electrodes Fz, F3, F4, FC3, FC4, Cz, C3 and C4.

For frequency-domain analysis of the alpha-band, we submitted the full epoch ranging between − 1.5 and 11.5 s to a wavelet convolution (wavelet width = 3 standard deviations of the implicit Gaussian kernel) that yielded complex output in frequency bins ranging between 8 and 12 Hz (in 0.5 Hz steps) and with 20 ms temporal resolution. Complex output at electrode Pz[43,73] was converted to an amplitude time series by taking the modulus and then averaging over frequency bands and trials. Alpha amplitude in the 1–10 s range was baseline corrected by subtracting and then dividing by alpha amplitude averaged over the − 1 to − 0.2 s time window. Note that trial resampling was not used here because we wanted to leave the alpha time courses intact. At each time point, a 2 (Age) × 2 (Attention) mixed-measures ANOVA was calculated. P values were corrected for multiple comparisons across time points separately for each main effect and the interaction using a false-discovery rate (FDR) correction[74,75]. Where the interaction reached significance, we tested for significance of attention effects separately for each age group using paired-samples t-tests at each time point, corrected across time points for multiple comparisons.

To examine the degree to which gap-detection performance was modulated by entrained pre-gap neural phase, we defined 3.9 s target epochs centred on the onset of each gap. First, the single-trial time-domain signal was detrended (using linear regression), then gap-evoked responses were removed from the post-gap-onset time window by multiplication with half of a Hann window that ranged between 0 and 50 ms post-gap-onset (and was zero thereafter). This was done in order to eliminate the possibility that evoked responses (or response differences between hit and miss trials) could be smeared back into the pre-stimulus period by the wavelet convolution and thereby causing spurious pre-gap phase effects[72,76]. Next, the time-domain data were submitted to a wavelet convolution (wavelet width = 3 standard deviations of the implicit Gaussian kernel) that yielded complex output in a single frequency bin centred on 2.8 Hz with 2-ms temporal resolution (that is, 500 Hz sampling frequency). Complex output was then converted to phase-angle time series. Single-trial phases averaged over fronto-central electrodes and in the time window ranging between − 0.024 and 0 s were sorted into 14 overlapping bins ranging between − π and + π (bin width = π/2, mean number of trials per bin = 60). Hit rate was calculated for each bin. The degree to which hit rates were modulated by neural phase (neuro-driven behavioural modulation) was quantified as the maximum minus the minimum hit rate. We tested whether neuro-driven behavioural modulation differed significantly between younger versus older listeners using a rank-sum test.

To bring the neural measures together with behaviour, we conducted several correlations. As outlined above, we predicted that stronger entrainment to FM sounds would lead to stronger modulation of hit rate by neural phase (that is, stronger neuro-driven behavioural modulation). For this reason, we correlated 2.8 Hz amplitude for the active condition from the FFT with neuro-driven behavioural modulation (that is, modulation of behaviour by 2.8 Hz neural phase). We also anticipated an effect of alpha amplitude (as an index of top-down attention) on behaviour. Thus, we correlated tonic alpha amplitude (averaged over time) from the active condition with neuro-driven behavioural modulation. Since these correlations were hypothesis-driven, significance was evaluated based on one-tailed P values. Across the board, we used nonparametric Spearman rank-order correlations due to the presence of an outlier in the younger group (more than 4 s.d. greater than the mean on 2.8-Hz amplitude). To preview, we

found a significant relation between alpha amplitude and neuro-driven behavioural modulation, but did not see the predicted entrainment-behaviour relationship. For that reason, we examined the inter-relations between entrainment, alpha, and behavioural modulation by correlating 2.8 Hz amplitude with neuro-driven behavioural modulation, partialing out alpha amplitude.

For analysis of ERPs, time-domain data from a fronto-central electrode cluster (Fz ,F3, F4, FC3, FC4, Cz, C3, C4) were low-pass filtered at 20 Hz (6th-order Butterworth) and averaged. ERPs were not baseline corrected since they were high-pass filtered to remove slow drifts (however, all results were consistent with those reported in the Results section when a baseline ranging from − 0.05 to 0 s before tone onset was subtracted from the ERPs). High-pass filtering instead of baseline correction is particularly well suited for fast presentation designs as employed in the present study[77,78]. Individual component amplitudes for the P1, N1 and P2 components were determined based on per-participant estimates of the latency of each component (individual latencies were constrained to fall within 30, 40 and 250 ms of the grand average peaks for the P1, N1 and P2, respectively). Latency estimates were confirmed by visual inspection and component amplitudes were then averaged within ± 10 ms of the individually determined peak. Components amplitudes were tested for age differences using independent-samples t-tests. Amplitudes in the P1, N1 and P2 components were also correlated with 2.8 Hz amplitude (both active and passive, also using nonparametric Spearman rank-order correlations; significance was evaluated based on two-tailed P-values for correlations with ERP amplitude, as we did not have specific hypotheses about relationships between entrainment and ERPs). P values were FDR-corrected across components, separately for correlations with active and passive entrainment.

We also conducted an analysis of the topographies of entrained neural responses as well as the topographies of individual ERP components. The rationale here was that neural responses generated by adjacent but not identical neural populations in auditory cortex should yield separable scalp topographies[79]. Thus, by comparing the topographies of ERP components to entrained neural responses, we were able to examine whether entrained responses share generators with obligatory sensory ERP components, and furthermore whether entrained neural responses might be explained as a superposition of evoked responses[52]. For active and passive entrained responses separately, and for the P1, N1 and P2 ERP components, we used a least-squares procedure to fit a two-dimensional Gaussian function with five free parameters to each topography: A (amplitude), $x_m$ (mean in left-right direction), $x_{s.d.}$ (standard deviation in left-right direction), $y_m$ (mean in anterior-posterior direction) and $y_{s.d.}$ (standard deviation in anterior-posterior direction). Statistical analyses were conducted on both parameters describing topographies in the anterior-posterior plane (that is, $y_m$ and $y_{s.d.}$). We first tested for differences between younger versus older participants for each type of neural response using separate nonparametric rank-sum tests (FDR-corrected across neural response types), and for active versus passive listening for entrained responses using a Wilcoxon sign-rank test. Then, we tested whether the entrainment topography (averaged over active and passive listening conditions) differed from the topography for each ERP-component (P1, N1, P2) for parameters $y_m$ and $y_{s.d.}$ estimated from the Gaussian fit. P values were FDR corrected separately for each parameter.

**Effect size.** Effect sizes for dependent and independent pairwise tests are reported as $r_{equivalent}$ (ref. 80) (throughout, r), which is equivalent to a Pearson product-moment correlation for two continuous variables. For nonparametric correlations, we report Spearman's rho, $\rho$, as the corresponding effect size measure. For ANOVAs, we report partial eta-squared, $\eta_P^2$.

**Data availability.** Raw data (EEG and behavioural data) are available by request from the corresponding author.

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

## Acknowledgements

The work was supported by a Max Planck Research Group grant from the Max Planck Society (to J.O.).

## Author contributions

M.J.H., B.H., D.K. and J.O. designed the experiment. D.K. collected the data. M.J.H. analysed the data. M.J.H., B.H., D.K. and J.O. wrote the paper.

## Additional information

**Competing interests:** The authors declare no competing financial interests.

