## [Peer Review File · Nature Communications]

Reviewers' comments:

Reviewer #1 (Remarks to the Author):

In this manuscript the authors provide an interesting approach to further understanding of age-related decreases in speech understanding. They used EEG to assess both sensory and cognitive changes with aging. They investigated neural entrainment at 2.8 Hz (a speech-relevant rate) to assess sensory-driven responses and alpha activity to assess cognitive top-down factors. The study is strengthened by an additional control condition using tone sequences to rule out the effects of age-related changes in general cortical responsiveness. The results are different for age groups and for type of oscillation. Older adults entrained less strongly than younger adults to the 2.8 Hz rhythm. Furthermore, whereas alpha activity increased during active performance in young adults, this activity decreased in older adults 3 sec into the task. Given that behavioral performance on the gap detection task did not differ between groups, these results suggest differing neural mechanism underlying behavioral performance between age groups. Knowledge of these mechanisms may lead to improved strategies for improving listening difficulties in older adults.

I just have a few comments/suggestions to improve clarity. I had some difficulty understanding the tasks and the measurements discussed in the introduction, especially. However, the figures are excellent, so things became clearer once I reached the results section.

The authors introduce the idea of stimulation-driven neural phase in the abstract and in the introduction. What is the motivation for examining this modulation in addition to the strength of neural entrainment? Also, it's hard to understand what the authors mean by stimulation-driven neural phase. It becomes clear when looking at Figure 1 and in the methods, but this concept is introduced without much explanation and I found it hard to follow.

I'm not sure what is meant by this sentence: "The results demonstrate that alpha oscillations play an insulating role, allowing the listener to ignore stimulus rhythm". Can the authors elaborate on "insulating role"?

What was the motivation for the additional time-domain analyses focused on the N1 component? This is clarified somewhat in the discussion but is presented without any lead-in statements in the Results.

Similarly, what is the motivation for the topography analysis?

Correlations with alpha. It would be interesting to see this by groups. The correlation between alpha and 2.8 Hz looks like it is driven by the younger group whereas the correlation between alpha and the hit rate looks like it is driven by the older group. A negative correlation in the older group is particularly interesting in that the alpha amplitude decreases with attention in the older group. If this negative correlation is present in only the older group, then the interpretation in the Discussion section may need to be modified.

In figure 7 it might be useful to include another diagram that shows relationships within groups, that is if the significance holds with a decrease in the N.

Finally, a recent study examined aging effects on neural synchronizations to speech-related acoustic modulations (Goossens et al., 2016). This study used amplitude modulations rather

than frequency modulations, but it may be worthwhile to refer to this study in the Discussion.

Goossens, T., Vercammen, C., Wouters, J., and van Wieringen, A. (2016). "Aging Affects Neural Synchronization to Speech-Related Acoustic Modulations," *Front Aging Neurosci* 8, 133.

Respectfully Submitted,
Samira Anderson, Au.D., Ph.D.

Reviewer #2 (Remarks to the Author):

This EEG paper investigates the oscillatory correlates/ substrates of sensory detection in an auditory task in young (18-31 yrs.) and elderly participants (61-77 yrs.). It dissociates between (a) bottom-up entrained oscillatory EEG signals (in response to a 2.8Hz amplitude modulated FM tone) that phase-aligns auditory detection performance to the input stream (serving a "rhythmic sampling mode") and (b) top-down modulated intrinsic alpha oscillations known to be related to detection of sensory signals through more sustained up-regulation (or down-regulation) of alpha-activity ("continuous sampling mode"). The results are interesting, as they indicate a flip in "sampling strategy" between young and older adults, leading to roughly identical performance levels. The data reveal that while older adults down-regulate alpha, younger adults upregulate this type of activity. The down-regulation in older participants seems to unmask (disinhibit) the modulation of detection performance by entrainment phase, and hence biases the sensory system into a "rhythmic sampling mode", as evidenced by (partial) correlation analysis between all measures (neuro-entrainment, behavioural entrainment, alpha-modulation). In contrast, up-regulation of alpha by younger participants masks (inhibits) the locking of behaviour to phase putatively switching to a "continuous sampling mode".

I am rather enthusiastic about this data set/ study. It is an exciting/ significant illustration of the existence of two sensory attentional sampling modes (Lakatos and Schroeder, *Science/TINS* 2008/2009). Here shown in human participants using detailed, sound (as far as I can judge) and up-to-date analysis. It is also very relevant as I think it can shed new light on some previous unexplained findings in the literature on distinct groups of alpha modulators (see below). My suggestions for improvements mostly concern framing of the findings, some addition of literature to intro/discussion and requests for clarifications and/or further analysis (mostly regarding the link between entrainment and ERPs, i.e. the control analysis).

Specific points

1) Framing

While the paper is very well written, I feel it does not put the most exciting aspects upfront. It took me quite some time to get into it and to understand the value of its contribution. It is only in the very last sentences of the discussion that I understood the important contribution it makes to the recent literature on sampling modes. I tried to emphasize the

points I am excited about in my summary above. I think the impact of the paper would improve by quite a bit if the authors would consider reframing it.

2) Literature bias

I feel that the literature review is not in all instances very well balanced. For instance, in the sections on the modulation of alpha activity by attention, the work of only two groups is cited (Foxe et al. et al., Jensen et al.), while others have been missed (Klimesch et al., Sauseng et al., Thut et al.). It would be nice the contribution of more groups could be acknowledged in the text.

3) Control analyses to rule out that the difference in entrained oscillations between young and old (active condition) does not reflect a difference in cortical excitability (based on ERP analysis)

I am not fully convinced that the above has been fully ruled out. The main argument relies on the enhancement of N1-amplitude in older vs. younger participants, which is in the opposite direction than the amplitude reduction in entrained oscillations in old vs young, and can therefore not explain it (argument against excitability changes explaining the entrainment results). However, there is also a significant ERP amplitude reduction in P2 which is not discussed at all. This is a problem as it reflects a reduction in excitability that may partially explain the reduction in entrained 2.8Hz oscillatory activity. Do these two amplitude changes (P2 vs entrained oscillations) correlate at all? An additional argument for the entrained oscillation to be unrelated to ERPs relies on apparent topographical differences between entrained activity and ERPs (P1, N1, P2). Here, again, this is not fully conclusive I think. While the topography of entrained activity is significantly different from the P1 and N1 topography, it is not significantly different from the P2-topography. This is not helping making the case as it is also the P2 which changes in amplitude in the direction of the entrained activity. In addition, I am wondering whether the comparisons between topographies (which seem to be based on the voltage gradients in only the anterior-posterior axis) is sensitive enough to capture all differences. Why not performing a whole-scalp (spatial) correlation analysis to probe dissimilarity between topographies. For an example, see e.g. Rihs et al., (2007). I think such an analysis will likely be more sensitive and may also reveal differences between the entrained activity and P2 in terms of topography (which would help).

Rihs TA, Michel CM, Thut G. Mechanisms of selective inhibition in visual spatial attention are indexed by alpha-band EEG synchronization. *Eur J Neurosci.* 2007 Jan;25(2):603-10.

4) I am finding this data set of interest also in light of some previous unexplained results in the literature on distinct groups of alpha modulators (Rihs et al., 2009). In Rihs et al (2009) a large portion of participants did not show an attentional related alpha modulation, while showing the same performance benefits from attention than the alpha-modulator group. Rihs et al (2009) have argued that different strategies between these participants may underlie this difference, although no specific strategy was put forth. It would be nice if the present result could be discussed in the light of these findings.

Rihs TA, Michel CM, Thut G. A bias for posterior alpha-band power suppression versus

enhancement during shifting versus maintenance of spatial attention. *Neuroimage*. 2009 Jan 1;44(1):190-9.

Reviewer #3 (Remarks to the Author):

This study examined age-related changes in temporal processing of auditory stimuli using EEG-based measures of neural entrainment and alpha oscillation amplitude, and phase. Younger and older adults identified brief gaps in frequency-modulated sounds. The results demonstrated age-related differences in neural entrainment at speech-relevant rates and in the amplitude of alpha oscillations for attended versus unattended stimuli. Different relationships were observed for these two measures in predicting gap detection performance. The authors conclude that the results provide evidence of the balance between different neural mechanisms for stimulus-driven entrainment and inhibition of task-irrelevant information during auditory perception.

The questions posed by the authors are novel and of interest to the field. The manuscript is of an appropriate length and well written. Most of my comments below suggest clarifying the motivation and predictions of the current study. (For context, I have expertise in neuroimaging measures of age-related changes in speech perception, but I am not an EEG expert.)

Major concerns

My primary recommendation is that the Introduction be expanded to better justify the study goals and predictions. (These issues were more clearly addressed in the Discussion.) Background literature is provided for age-related changes in neural entrainment and alpha oscillations broadly. However, the theoretical justification for examining alpha phase and amplitude specifically could be strengthened (e.g., amplitude is briefly mentioned as an index of top-down inhibition in last paragraph without citations). For non-EEG experts, this would aid our understanding the predictions regarding the link between these measures and entrainment.

I interpreted the aim of this manuscript to be the investigation of the relative contribution of sensory-driven and top-down age-related changes to auditory processing. I recommend strengthening this narrative throughout the manuscript, particularly given the number of measures/results. Perhaps consider:

1) expanding Intro predictions by including alternative hypotheses (e.g., Is aging predicted to equally affect sensory/top-down factors? What does it mean if the effect is on primarily one or the other?) The authors address such topics in the Discussion, but a preview is missing from the Intro.

2) adjusting Result headings to highlight the theoretical question rather than the analytical approach (e.g., as simple as adding "sensory-driven" or "top-down" to the headings, or grouping them under higher-level headings that broadly restate the questions in the Intro).

Greater discussion of the study's limitations/future directions seems warranted. For example, the authors describe a negative correlation between amplitude and behavior as evidence of inhibition of rhythmic information (p. 15). Might an appropriate future direction be to examine this relationship nearer to the trial-level rather than on an individual subject basis? Do the results of the study yield predictions about the impact of hearing loss on the neural encoding of speech information (e.g., impact on sensory-driven neural entrainment)?

Minor concerns

Please add a statement justifying the selection of the 2.8 Hz rate and its "speech-relevance."

Please include additional descriptive details about the participants in the methods section (e.g., sex, age M and SD, study exclusion criteria).

Are the units in the audiogram plots in Fig. 1B meant to be in dB HL, instead of SPL? If possible, state explicitly that the 50 dB SPL presentation exceeds the hearing thresholds for each participant at the critical stimuli frequencies (i.e., audibility was ensured). Given the equivalence in behavioral performance across age groups, I assume this is the case.

In the instructions for the passive condition, were participants actually told to "ignore" the stimuli (i.e., suggesting an active, intentional process) or rather told that they do not have to respond to the stimuli? "Passive listening" and "ignoring" seem different to me, unless this is the terminology commonly used with this type of design.

Was there a lower bound cutoff off for reaction times (e.g., RTs < 200ms might be late responses to a previous stimulus)?

The authors may want to consider citing one of these recent papers as additional evidence of age-related cortical over-representation in longer duration speech samples (p. 12), perhaps in the context the implications of the current work for naturalistic speech processing:

Presacco, A., Simon, J. Z., & Anderson, S. (in press). Evidence of degraded representation of speech in noise, in the aging midbrain and cortex. *Journal of Neurophysiology*. doi:10.1152/jn.00372.2016

Presacco, A., Simon, J. Z., & Anderson, S. (in press). Effect of informational content of noise on speech representation in the aging midbrain and cortex. *Journal of Neurophysiology*. doi:10.1152/jn.00373.2016

Reviewer #4 (Remarks to the Author):

Henry and colleagues in their paper „The balance between rhythmic neural entrainment and top-down neural modulation changes with age“ study the ability of healthy younger and

older subjects to detect short gaps in sound trains. For this purpose, they use a paradigm very similar to their previous study (Henry and Obleser 2012), in which they demonstrated that the successful detection of the gap depends on the delta phase at the CZ electrode. In the present paper, they now compare differences in entrainment frequency and alpha amplitude between young and older subjects. The major finding is that the older subjects suppress the alpha oscillations more than the younger subjects, while the entrainment is less pronounced in the older subjects. The comparison of older and younger subjects is intriguing in trying to explain why older subjects have more difficulties in separating multiple sounds. These results are of interest for researchers studying the auditory domain and might also have interesting implications for studies on ageing.

While the paper is well written, the methods description is rather on the short side, leaving many technical questions open that are crucial for judging the validity of the results. For that reason, some of the results are not convincing without further analysis and explanation. For example, clear information from which EEG electrode the different results are coming from at each step is missing and makes the paper rather confusing to read.

Additionally, the paper and its claims would have greatly benefited from using high-density EEG or MEG and then performing the analysis on the cortical surface. Such an analysis would allow identifying neural generators, which the authors discuss, but cannot pinpoint down. I understand that this would require a complete new experiment and is therefore most likely beyond the scope of the present paper. The low resolution and the analysis in the sensor space nevertheless negatively impact the value added for the research community.

I will detail my concerns in the following:

Major Points:

1) Figure 3 and corresponding methods: Which baseline was used? The baseline – if it is the one plotted in the ERP plots – is much lower for the younger subjects. This could largely explain the effect of the amplitude differences in young and older subjects. Therefore at the moment I cannot judge whether the described ERP effect is a true one or related to an inappropriate baseline choice.

Some further labels on the ERP plots would be very helpful as well as on the small bar graph inlays. Which electrodes were included in the ERP analysis?

2) For the phase-estimation it is essential to see that the results are unaffected by the Wavelet filtering (Zoefel and Heil 2013). Even though the ERP is removed, I still would be more convinced that there is a true phase effect if the analysis would be also done with an acausal filter.

3) Regarding the result of no correlation between hit rate and neural amplitude for both subject groups together: How can this be explained in relation to the previous results in young subjects (Henry and Obleser 2012), where a correlation was present. I understand that when controlling for alpha amplitude a partial correlation is found, but that was not necessary in the previous study. Therefore, the results seem to contradict each other.

4) On page 8 2 subjects are excluded from analysis. From which group were the 2 subjects excluded? Was the 1 subject that was excluded from the later analysis among them? Were the 2 subjects included in the following analysis?

Minor Points:

5) Page 5: Stimulus-driven behavioral modulation: Is this the measure described on page 19? Here it is not clear which particular kind of smoothing kernel is taken and how this

choice is justified. At the same time, these are only behavioral data for which the minimum and maximum are taken so that smoothing seems unnecessary. In Figure 1c the y-axis labels are missing. Particularly for the last plot it is not obvious what is plotted there.

6) Figure 2B: Please specify for which frequency the amplitudes are taken (2.8 Hz?). I guess that the amplitude is calculated from the fronto-central electrodes described on page 21.

7) Figure 4: A y-axis would be helpful.

8) Page 9: Which Figure is meant by 3B? There is no figure 3B.

9) Figure 5 -7: The measure "hit rate (peak-trough)" is not explained. Is it the stimulus-driven behavior modulation described on page 19? Also the residual of this cannot be understood without further explanation. Units on the y-axis are missing.

10) Figure 7: Across which electrodes was the alpha amplitude taken? Is it Pz as mentioned in the methods section?

11) Please also provide the mean age and standard deviation besides the range.

12) Page 17: SL has not been introduced.

13) Page 22: Why are 14 phase bins chosen for the neuro-driven behavioral modulation while 20 bins are chosen for the stimulus driven behavioral modulation? For comparison the same number of bins would be more appropriate.

Reviewer #1 (Remarks to the Author):

In this manuscript the authors provide an interesting approach to further understanding of age-related decreases in speech understanding. They used EEG to assess both sensory and cognitive changes with aging. They investigated neural entrainment at 2.8 Hz (a speech-relevant rate) to assess sensory-driven responses and alpha activity to assess cognitive top-down factors. The study is strengthened by an additional control condition using tone sequences to rule out the effects of age-related changes in general cortical responsiveness. The results are different for age groups and for type of oscillation. Older adults entrained less strongly than younger adults to the 2.8 Hz rhythm. Furthermore, whereas alpha activity increased during active performance in young adults, this activity decreased in older adults 3 sec into the task. Given that behavioral performance on the gap detection task did not differ between groups, these results suggest differing neural mechanism underlying behavioral performance between age groups. Knowledge of these mechanisms may lead to improved strategies for improving listening difficulties in older adults.

I just have a few comments/suggestions to improve clarity. I had some difficulty understanding the tasks and the measurements discussed in the introduction, especially. However, the figures are excellent, so things became clearer once I reached the results section.

Thanks to Prof. Anderson for the positive and constructive review. We hope that the major revision we've undertaken substantially improves the clarity of the manuscript.

The authors introduce the idea of stimulation-driven neural phase in the abstract and in the introduction. What is the motivation for examining this modulation in addition to the strength of neural entrainment? Also, it's hard to understand what the authors mean by stimulation-driven neural phase. It becomes clear when looking at Figure 1 and in the methods, but this concept is introduced without much explanation and I found it hard to follow.

Our intention was to make clear the distinction between neural oscillations that are responding directly to the temporal structure of a stimulus (entrained neural oscillations) and neural oscillations that are more internally driven and reflect task demands (alpha oscillations). We agree though that the terms 'stimulus-driven neural phase' and 'stimulus-driven neural entrainment' are quite confusing, especially early on before the stimuli and paradigm are described. We have revised the Abstract and Introduction to remove these terms, and now refer to 'neural entrainment' or 'entrained neural phase' throughout. We have also added a brief description of the task to the Introduction, so that hypotheses regarding neural entrainment will have a more solid foundation.

I'm not sure what is meant by this sentence: "The results demonstrate that alpha oscillations play an insulating role, allowing the listener to ignore stimulus rhythm". Can the authors elaborate on "insulating role?"

We have changed this wording now to avoid the term "insulating". We do still talk about "shielding", which is very similar, but we've tried to clarify what we mean by that.

p. 5, Introduction: "The results demonstrate that alpha oscillations effectively 'shield' the listening brain from obligatory behavioral entrainment by the stimulus rhythm: In keeping with a listener's

engagement in continuous-mode processing, higher alpha amplitude was associated with a reduced degree of behavioral modulation by entrained neural phase.”

p. 18, Discussion, Dynamics of top-down neural modulation change with age: “Here, we suggest that enhanced alpha amplitude during task performance reflects selective inhibition of the rhythmic information conveyed by the frequency modulation, in an effort to keep performance consistently high over time (referred to above as a ‘shielding from entrainment’ role of alpha). This hypothesis is supported by the negative correlation between alpha amplitude and neuro-driven behavioral modulation – the higher a participant’s alpha amplitude, the weaker the influence of rhythm on gap-detection performance.”

What was the motivation for the additional time-domain analyses focused on the N1 component? This is clarified somewhat in the discussion but is presented without any lead-in statements in the Results.

We have removed these analyses from the revised manuscript.

Similarly, what is the motivation for the topography analysis?

We have added the following motivation to the Results section:

pp. 9–10, Results, Generators of entrained neural responses are discriminable from auditory ERPs: “The analysis of topographical distributions allowed us to test whether the generators of neural responses that were entrained at 2.8 Hz would be dissociable from the generators of ERPs evoked by tones presented at 2.8 Hz. This analysis speaks to the ongoing debate in the literature as to whether peaks in EEG spectra in response to modulation reflect entrainment of ongoing (spontaneous) neural oscillations⁵⁴⁻⁵⁷ or whether the same spectral peaks might instead reflect the summation of a series of invariant transient neural responses evoked by rhythmic stimulation independent of ongoing oscillatory activity⁵⁸.”

p. 11, Results, Generators of entrained neural responses are discriminable from auditory ERPs: “Dissociation of the topographical distributions of the two response types provides evidence against the assertion that spectral peaks would necessarily reflect only a series of evoked responses.”

Correlations with alpha. It would be interesting to see this by groups. The correlation between alpha and 2.8 Hz looks like it is driven by the younger group whereas the correlation between alpha and the hit rate looks like it is driven by the older group. A negative correlation in the older group is particularly interesting in that the alpha amplitude decreases with attention in the older group. If this negative correlation is present in only the older group, then the interpretation in the Discussion section may need to be modified.

In figure 7 it might be useful to include another diagram that shows relationships within groups, that is if the significance holds with a decrease in the N.

We evaluated pairwise correlations between alpha amplitude and neuro-driven behavioral modulation as well as alpha amplitude and entrainment strength separately for the two age groups. We’ve done

the same for the partial correlation between entrainment strength and neuro-driven behavioral modulation (with alpha partialled out).

As intuited by the Reviewer, when tested separately for the two age groups, the correlation between alpha amplitude and neuro-driven behavioral modulation was nonsignificant for younger adults ($\rho = -.32$, $p = .08$, though arguably marginally so), but significant for older adults ($\rho = -.49$, $p = .01$). However, the correlations were not significantly different from each other ($z = 0.6$, $p = .55$).

The direct correlation between alpha amplitude and entrainment strength was significant for both age group separately (younger: $\rho = .44$, $p = .03$; older: $\rho = .52$, $p = .01$), and did not differ significantly between groups ($z = -.30$, $p = .76$).

Finally, when we considered the age groups separately, the partial correlation between entrainment strength and neuro-driven behavioral modulation was significant for the young ($\rho = .42$, $p = .04$) and was marginally so for the older group ($\rho = .34$, $p = .07$). The correlation coefficients were not significantly different between groups ($z = 0.27$, $p = .79$).

As suggested by the Reviewer, we now provide best-fit lines separately for the younger and older groups in Figure 7 specifically. However, given the lack of significant differences between groups in terms of correlation magnitude for any relationship and the stronger consistency across groups using nonparametric Spearman correlations, we want to be careful not to over-interpret any visual differences. We have opted not to include correlation coefficients separately for each group plus tests of differences between groups for each pair of measures, as we don't want to dilute the main findings. We hope that the Reviewer supports this decision.

Finally, a recent study examined aging effects on neural synchronizations to speech-related acoustic modulations (Goossens et al., 2016). This study used amplitude modulations rather than frequency modulations, but it may be worthwhile to refer to this study in the Discussion.

Goossens, T., Vercammen, C., Wouters, J., and van Wieringen, A. (2016). "Aging Affects Neural Synchronization to Speech-Related Acoustic Modulations," Front Aging Neurosci 8, 133.

Thank you very much for pointing us to this new paper – we missed it before submitting our own. We now refer to the study in the Discussion, and have additionally cited it in a discussion of the importance of taking into account modulation rate when comparing age groups.

p. 15, Discussion, Entrainment to frequency modulation becomes weaker and less flexible with age: "A similar age-related increase in neural responses to amplitude-modulated stimuli was recently reported for a 4-Hz modulation rate and 100% modulation depth⁶⁴."

p. 16, Discussion, Entrainment to frequency modulation becomes weaker and less flexible with age: "Several factors have been shown to modulate age effects on neural entrainment, including the presence⁶¹ and content⁶⁵ of background noise as well as the rate^{26,52,64,68} and depth⁶³ of the stimulus modulation."

Respectfully Submitted,
Samira Anderson, Au.D., Ph.D.

Reviewer #2 (Remarks to the Author):

This EEG paper investigates the oscillatory correlates/ substrates of sensory detection in an auditory task in young (18-31 yrs.) and elderly participants (61-77 yrs.). It dissociates between (a) bottom-up entrained oscillatory EEG signals (in response to a 2.8Hz amplitude modulated FM tone) that phase-aligns auditory detection performance to the input stream (serving a “rhythmic sampling mode”) and (b) top-down modulated intrinsic alpha oscillations known to be related to detection of sensory signals through more sustained up-regulation (or down-regulation) of alpha-activity (“continuous sampling mode”). The results are interesting, as they indicate a flip in “sampling strategy” between young and older adults, leading to roughly identical performance levels. The data reveal that while older adults down-regulate alpha, younger adults upregulate this type of activity. The down-regulation in older participants seems to unmask (disinhibit) the modulation of detection performance by entrainment phase, and hence biases the sensory system into a “rhythmic sampling mode”, as evidenced by (partial) correlation analysis between all measures (neuro-entrainment, behavioural entrainment, alpha-modulation). In contrast, up-regulation of alpha by younger participants masks (inhibits) the locking of behaviour to phase putatively switching to a “continuous sampling mode”.

I am rather enthusiastic about this data set/ study. It is an exciting/ significant illustration of the existence of two sensory attentional sampling modes (Lakatos and Schroeder, Science/TINS 2008/2009). Here shown in human participants using detailed, sound (as far as I can judge) and up-to-date analysis. It is also very relevant as I think it can shed new light on some previous unexplained findings in the literature on distinct groups of alpha modulators (see below). My suggestions for improvements mostly concern framing of the findings, some addition of literature to intro/discussion and requests for clarifications and/or further analysis (mostly regarding the link between entrainment and ERPs, i.e. the control analysis).

We’d like to thank the Reviewer for the enthusiastic review of our manuscript, and for great suggestions that we think have substantially improved the manuscript’s quality.

Specific points

1) Framing

While the paper is very well written, I feel it does not put the most exciting aspects upfront. It took me quite some time to get into it and to understand the value of its contribution. It is only in the very last sentences of the discussion that I understood the important contribution it makes to the recent literature on sampling modes. I tried to emphasize the points I am excited about in my summary above. I think the impact of the paper would improve by quite a bit if the authors would consider reframing it.

We’ve now rewritten the Introduction and some of the Discussion, taking into account this Reviewer’s important comment in combination with a similar comment from Reviewer #3. We think that this has strengthened the paper a lot, and hope that the Reviewer agrees with our re-framing.

2) Literature bias

I feel that the literature review is not in all instances very well balanced. For instance, in the sections on the modulation of alpha activity by attention, the work of only two groups is cited (Foxe et al. et al., Jensen et al.), while others have been missed (Klimesch et al., Sauseng et al., Thut et al.). It would be nice the contribution of more groups could be acknowledged in the text.

Thanks for this. We have added and substituted references to give a more global, balanced view of the existing literature on alpha oscillations.

3) Control analyses to rule out that the difference in entrained oscillations between young and old (active condition) does not reflect a difference in cortical excitability (based on ERP analysis)

I am not fully convinced that the above has been fully ruled out. The main argument relies on the enhancement of N1-amplitude in older vs. younger participants, which is in the opposite direction than the amplitude reduction in entrained oscillations in old vs young, and can therefore not explain it (argument against excitability changes explaining the entrainment results). However, there is also a significant ERP amplitude reduction in P2 which is not discussed at all. This is a problem as it reflects a reduction in excitability that may partially explain the reduction in entrained 2.8Hz oscillatory activity. Do these two amplitude changes (P2 vs entrained oscillations) correlate at all? An additional argument for the entrained oscillation to be unrelated to ERPs relies on apparent topographical differences between entrained activity and ERPs (P1, N1, P2). Here, again, this is not fully conclusive I think. While the topography of entrained activity is significantly different from the P1 and N1 topography, it is not significantly different from the P2-topography. This is not helping making the case as it is also the P2 which changes in amplitude in the direction of the entrained activity. In addition, I am wondering whether the comparisons between topographies (which seem to be based on the voltage gradients in only the anterior-posterior axis) is sensitive enough to capture all differences. Why not performing a whole-scalp (spatial) correlation analysis to probe dissimilarity between topographies. For an example, see e.g. Rihs et al., (2007). I think such an analysis will likely be more sensitive and may also reveal differences between the entrained activity and P2 in terms of topography (which would help).

Rihs TA, Michel CM, Thut G. Mechanisms of selective inhibition in visual spatial attention are indexed by alpha-band EEG synchronization. Eur J Neurosci. 2007 Jan;25(2):603-10.

These are very important points. In response to this comment, we now more fully explore the P2 ERP effects and their relation to age-related entrainment differences. In fact, there was a strong and significant correlation between P2 amplitude and 2.8-Hz amplitude, but only for the active condition. Interestingly, ERP P2 amplitude was not correlated with entrainment strength (2.8-Hz amplitude) measured during passive listening, even though ERPs were actually measured during passive listening. We have added these analyses and discuss the findings in the manuscript's Discussion. In short, we think the worry that P2 and entrainment stem from the same reduced sensitivity is unlikely for two reasons: first, the entrainment–P2 correlation only holds for active, but not for passive listening; and 2) the topographies for P2 and entrainment are dissociable (described in more detail below).

With respect to the topography analysis, we appreciate the Reviewer's suggestion to further hunt down differences between the entrainment topography and the ERP topographies. We experimented

with the suggested spatial correlation analysis, but ultimately had trouble settling on a dependent measure that clearly, statistically, captured what the average topographies clearly show.

However, in exploring the data further, we realized that the parameter estimates from the Gaussian fits were non-normally distributed (something we had overlooked before). Testing the parameter estimates using appropriate nonparametric tests revealed that the entrainment topography was statistically dissociable from all ERP-component topographies for the anterior-poster center parameter, as well as the standard deviation parameter, which we have added to the revised manuscript.

Relevant text and figures are reproduced below for the Reviewer's convenience.

pp. 8–9, Results, Entrainment differences are not due to differences in overall cortical responsiveness: “ERPs evoked by individual tones in sequences that had presentation rates and spectral ranges matched to the frequency-modulated sounds were examined to rule out the possibility that entrainment differences between younger and older adults could be due to a generalized reduction in cortical responsiveness to sound in older adults^{25,53}. A comparison between age groups for ERP amplitudes in the range of the P1, N1, and P2 components revealed significant differences for the N1 and P2 (P1: $t(38) = -1.63$, $p = .11$; N1: $t(38) = 3.21$, $p = .003$; P2: $t(38) = 2.25$, $p = .03$; Figure 3A). In particular, the N1 component was larger for older adults while the P2 component was smaller.

Since the direction of the age effect on the P2 was the same as the age effect on entrainment strength, we correlated P2 amplitudes with active and passive entrained 2.8-Hz amplitudes (we did the same for P1 and N1 amplitudes and found no significant relationships, all $pFDR \geq .31$). P2 amplitudes were significantly correlated with 2.8-Hz amplitude in the active ($\rho = .49$, $pFDR = .005$; Fig. 3B), but not the passive condition ($\rho = .14$, $pFDR = .58$; Fig. 3B). Thus, although N1 amplitudes were actually larger for older compared to younger adults^{53,54}, P2 amplitudes were reduced in older adults and correlated with the magnitude of 2.8-Hz entrainment. This held also for separate analyses in younger ($\rho = .54$, $p = .02$) and in older adults ($\rho = .43$, $p = .06$), ruling out the possibility that the overall correlation reflects age effects on both variables.

Interestingly though, this correlation was present only for active entrainment, and not for the passive condition, despite P2s being measured under passive listening conditions as well. In an effort to better characterize whether entrained neural responses and P2s may have arisen from the same neural generators, we performed an in-depth analysis of topographies of different neural response types.”

Figure 3. ERPs in response to individual tones. **A)** Older adults showed enhanced N1 responses and reduced P2 responses relative to younger adults. Individual lines show ERPs to each tone frequency. **B)** P2 amplitude was significantly correlated with 2.8-Hz amplitude measured during active task performance (left), but not with 2.8-Hz amplitude measured during passive listening (right). Insets show plots of ranks on which Spearman correlations were based. Ranks on 2.8-Hz amplitude are plotted on the x-axis and P2 amplitude on the y-axis.

pp. 9–10, Results, Generators of entrained neural responses are discriminable from auditory ERP generators: “We also compared topographies of the entrained 2.8-Hz neural response to ERP-component topographies for the P1, N1, and P2 (Fig. 4). The analysis of topographical distributions allowed us to test whether the generators of neural responses that were entrained at 2.8 Hz would be dissociable from the generators of ERPs evoked by tones presented at 2.8 Hz. This analysis speaks to the ongoing debate in the literature as to whether peaks in EEG spectra in response to modulation reflect entrainment of ongoing (spontaneous) neural oscillations⁵⁵⁻⁵⁸ or whether the same spectral peaks might instead reflect the summation of a series of invariant transient neural responses evoked by rhythmic stimulation independent of ongoing oscillatory activity⁵⁹.

First, we tested whether age affected topographies for any neural response type. We submitted anterior–posterior topography centers and their standard deviations (y_m and y_{sd} , respectively), quantified via two-dimensional Gaussian fits, to separate nonparametric rank-sum tests (FDR-corrected across neural response types). None of the tests revealed an age effect (all $pFDR \geq .09$). Next, we confirmed that attention did not significantly affect the topography for the 2.8-Hz entrained neural responses (active versus passive) using separate Wilcoxon sign-rank tests for each model parameter (FDR-corrected across parameters), all $pFDR \geq .25$. For subsequent statistical comparisons of neural responses, we collapsed across age and attention conditions.

Separate Wilcoxon sign-rank tests for 2.8-Hz topography versus each ERP-component topography were conducted for the fitted Gaussian parameters. The entrained 2.8-Hz topography was dissociable from ERP topographies based on both Gaussian parameters describing topographies in the anterior–posterior plane. The entrainment topography differed from all ERP-component topographies in terms of both center, y_m , (all $pFDR \leq .04$) and standard deviation, y_{sd} , (all $pFDR \leq .0001$). Thus the entrainment topography was centered less anterior and spread wider in the anterior–posterior direction than any of the ERP topographies. Dissociation of the topographical distributions of the two

response types provides evidence against the assertion that spectral peaks would necessarily reflect only a series of evoked responses.”

Figure 4. Topographies for entrained neural responses (2.8-Hz) differ from ERP component topographies (P1, N1, P2). Actual topographies for each neural response type are shown in the center of the plot (arranged horizontally, 2.8-Hz, P1, N1, and P2, responses; note that for this analysis the N1 topography was sign-inverted). Bar graphs (top) summarize the median (bar) and single-participant (black dots) data for each type of neural response and each fitted Gaussian parameter (y_m on left and y_{sd} on right). Note: Two outliers are not shown for the 2.8-Hz condition for parameter y_{sd} (values were 8.15 and 40.24). Line plot (bottom) shows predicted topographies for each neural response type in the anterior-posterior plane (right) based on median parameter values over participants. Entrained-response topographies were dissociable from ERP-component topographies.

pp. 29–30, Methods, Data acquisition and analysis, ERP and topography analysis: “Statistical analyses were conducted on both parameters describing topographies in the anterior-posterior plane (i.e., y_m and y_s). We first tested for differences between younger versus older participants for each type of neural response using separate nonparametric rank-sum tests (FDR-corrected across neural response types), and for active versus passive listening for entrained responses using a Wilcoxon sign-rank test. Then, we tested whether the entrainment topography (averaged over active and passive listening conditions) differed from the topography for each ERP component (P1, N1, P2) for parameters y_m and y_s estimated from the Gaussian fit. P-values were FDR corrected separately for parameter y_m and y_s .”

4) I am finding this data set of interest also in light of some previous unexplained results in the literature on distinct groups of alpha modulators (Rihs et al., 2009). In Rihs et al (2009) a large portion of participants did not show an attentional related alpha modulation, while showing the same performance benefits from attention than the alpha-modulator group. Rihs et al (2009) have argued that different strategies between these participants may underlie this difference, although no specific strategy was put forth. It would be nice if the present result could be discussed in the light of these findings.

Rihs TA, Michel CM, Thut G. A bias for posterior alpha-band power suppression versus enhancement during shifting versus maintenance of spatial attention. *Neuroimage*. 2009 Jan 1;44(1):190-9.

Thanks to the Reviewer for pointing out this interesting and very relevant paper. We have now added a brief discussion of these results into our Discussion in the context of considering age-related differences in neural strategy.

p. 18, Discussion, Dynamics of top-down neural modulation change with age: "This suggestion is particularly interesting in light of evidence that different strategies for shifting and maintaining attention can be observed even in young participants – in particular, whether or not an individual's alpha power was modulated in anticipation of a visual target was predictable from their resting alpha-power level⁷⁵. Although we did not test for age differences in resting alpha power in the current study, it is possible that different strategies used by younger and older listeners may have been in part predictable from resting alpha levels."

Reviewer #3 (Remarks to the Author):

This study examined age-related changes in temporal processing of auditory stimuli using EEG-based measures of neural entrainment and alpha oscillation amplitude, and phase. Younger and older adults identified brief gaps in frequency-modulated sounds. The results demonstrated age-related differences in neural entrainment at speech-relevant rates and in the amplitude of alpha oscillations for attended versus unattended stimuli. Different relationships were observed for these two measures in predicting gap detection performance. The authors conclude that the results provide evidence of the balance between different neural mechanisms for stimulus-driven entrainment and inhibition of task-irrelevant information during auditory perception.

The questions posed by the authors are novel and of interest to the field. The manuscript is of an appropriate length and well written. Most of my comments below suggest clarifying the motivation and predictions of the current study. (For context, I have expertise in neuroimaging measures of age-related changes in speech perception, but I am not an EEG expert.)

We thank the Reviewer for her/his suggestions regarding framing. We have rewritten the Introduction and much the Discussion, included a more detailed description of hypothesis in the Intro, and added a Future Directions / Limitations section to the Discussion. We hope that these revisions have improved the quality of our manuscript.

Major concerns

My primary recommendation is that the Introduction be expanded to better justify the study goals and predictions. (These issues were more clearly addressed in the Discussion.) Background literature is provided for age-related changes in neural entrainment and alpha oscillations broadly. However, the theoretical justification for examining alpha phase and amplitude specifically could be strengthened (e.g., amplitude is briefly mentioned as an index of top-down inhibition in last paragraph without citations). For non-EEG experts, this would aid our understanding the predictions regarding the link between these measures and entrainment.

We've now rewritten the Introduction and some of the Discussion, taking into account this Reviewer's important comment in combination with a similar comment from Reviewer #2. We have added explicit hypotheses to the Introduction. We think that this has strengthened the paper a lot, and hope that the Reviewer agrees with our re-framing.

I interpreted the aim of this manuscript to be the investigation of the relative contribution of sensory-driven and top-down age-related changes to auditory processing. I recommend strengthening this narrative throughout the manuscript, particularly given the number of measures/results. Perhaps consider:

1) expanding Intro predictions by including alternative hypotheses (e.g., Is aging predicted to equally affect sensory/top-down factors? What does it mean if the effect is on primarily one or the other?) The authors address such topics in the Discussion, but a preview is missing from the Intro.

We have expanded our Introduction and include explicit hypotheses (and motivations) for each dependent measure.

2) adjusting Result headings to highlight the theoretical question rather than the analytical approach (e.g., as simple as adding "sensory-driven" or "top-down" to the headings, or grouping them under higher-level headings that broadly restate the questions in the Intro).

Thanks for this suggestion. We have adjusted our Results headings to better reflect the theoretical/conceptual contributions rather than the specific analytical approaches.

Greater discussion of the study's limitations/future directions seems warranted. For example, the authors describe a negative correlation between amplitude and behavior as evidence of inhibition of rhythmic information (p. 15). Might an appropriate future direction be to examine this relationship nearer to the trial-level rather than on an individual subject basis? Do the results of the study yield predictions about the impact of hearing loss on the neural encoding of speech information (e.g., impact on sensory-driven neural entrainment)?

This is a great idea, and exploring this on a refined, individual-subject basis is indeed a logical and necessary next step. We have added a subsection to the Discussion describing the limitations / future directions of the current study. We have reproduced that subsection below.

pp. 19–20, Discussion, Future directions and limitations: "The current study aimed to test the hypothesis that sensory-driven neural entrainment, as well as top-down neural modulation, changes with age. The motivation was that age-related changes in either measure might contribute to listening and speech comprehension difficulties, and in particular difficulty solving the cocktail-party problem. However, the current study neither makes use of naturalistic speech stimuli^{63,64}, nor does it attempt to recreate the cocktail-party problem by including distractor sounds. Thus, one important future direction is to extend this work using stimuli and paradigms that better approximate the naturalistic sounds and situations that cause problems for older adults (e.g., competing speech, background noise, time-compressed speech). Using such manipulations to create a situation in which older adults cannot

default to a rhythmic-mode strategy when a top-down, alpha-based strategy fails might clarify the contributions to age-related speech comprehension deficits.

Moreover the time-scale of the effects discussed in the current study is relatively gross; we have aggregated over all trials to observe a trade-off between the effects of entrainment and alpha amplitude on behavior. However, a more fine-grained examination of trial-by-trial trade-offs between measures could reveal more subtle dynamical shifts between strategies or processing modes. Such an analysis requires a different behavioral measure than neuro-driven behavioral modulation, which we focused on here, since this measure requires aggregation over many trials to calculate.

Finally, an important future direction is to examine the impact of hearing loss (more severe than in our older sample) on neural entrainment to speech rhythm and the relationship of entrainment to speech comprehension. Hearing loss changes the frequency content arriving at the cortex, and manipulations such as noise vocoding that degrade acoustics result in reductions in entrainment strength^{20,21}. Moreover, a leading theory of hearing loss and cognition suggests that cognitive deficits can stem from hearing loss^{7,8}; thus it's possible that the balance between sensory-driven neural entrainment and top-down neural modulation is further shifted with moderate-to-severe hearing loss."

Minor concerns

Please add a statement justifying the selection of the 2.8 Hz rate and its "speech-relevance."

We have added this information to the Introduction.

p. 4, Introduction: "A frequency modulation rate of 2.8 Hz was chosen because this rate is representative of frequency fluctuations in natural speech corresponding to intonation contour⁵⁰."

Please include additional descriptive details about the participants in the methods section (e.g., sex, age M and SD, study exclusion criteria).

We have added this information to the Methods section.

p. 22, Methods, Participants: "Forty individuals (20 younger [10 male, 10 female], age 18–31 years, M = 25.4 years \pm SD = 3.3 years; 20 older [8 male, 12 female], age 61–77 years, M = 67.3 years \pm SD = 5.3 years) took part in the experiment."

Are the units in the audiogram plots in Fig. 1B meant to be in dB HL, instead of SPL? If possible, state explicitly that the 50 dB SPL presentation exceeds the hearing thresholds for each participant at the critical stimuli frequencies (i.e., audibility was ensured). Given the equivalence in behavioral performance across age groups, I assume this is the case.

The units in Figure 1B were indeed meant to be in dB HL; thanks for catching that. We've updated the figure.

Thanks for this recommendation re: stating ensured audibility for all participants. We have added the following text to the Methods section:

p. 22, Methods, Stimuli: "All stimuli were normalized with respect to peak amplitude, and were presented at 50 dB sensation level (SL; hearing thresholds were determined prior to EEG recordings for

each participant individually for a 1200-Hz sine tone using the method of limits and all stimuli were presented 50 dB above the individual hearing threshold to ensure audibility)."

In the instructions for the passive condition, were participants actually told to "ignore" the stimuli (i.e., suggesting an active, intentional process) or rather told that they do not have to respond to the stimuli? "Passive listening" and "ignoring" seem different to me, unless this is the terminology commonly used with this type of design.

Participants were indeed told to ignore the stimuli. Most studies measuring auditory steady-state responses to frequency or amplitude modulation are "passive", and although the specific instructions to the participant are usually not provided in the Methods, it is often the case that participants watch a silent movie with subtitles during presentation of the auditory stimuli. We suspect that they are actually ignoring the stimuli in such a situation. We of course can't be certain about this though.

We have added a note about this to the Methods section:

p. 23, Methods, Procedure: "(We note that explicitly instructing participants to ignore the auditory stimuli potentially departs from a classic definition of "passive" listening. However, many studies measuring neural responses to frequency and amplitude modulation during "passive" listening allow participants to watch a silent movie, sometimes with subtitles^{54,77,78}, so it's likely that these situations also involve ignoring.)"

Was there a lower bound cutoff off for reaction times (e.g., RTs < 200ms might be late responses to a previous stimulus)?

We did not use a lower bound cutoff for reaction times (RTs). However, the minimum duration separating individual gaps was 1.5 s, so any short RT that was actually a delayed response to the previous stimulus would have had to exceed 1.5 s.

In order to rule out the possibility that we were considering late reactions to earlier gaps as "hits" to gaps in a different phase position, we reanalyzed the behavioral data with a lower bound cutoff (200 ms). We did two things. First, we counted the number of responses that were excluded with the cutoff that had been included in our original analysis. Second, we correlated hit rates and stimulus-driven behavioral modulation scores for this new analysis with the measures from our original analysis.

The median number of excluded trials based on a lower-bound reaction time cutoff was 1 ± 1 (sIQR). A histogram of the number of excluded trials (Fig. R1) shows that all but 2 participants had a discrepancy of 5 trials or less (2% of total trials) with and without a lower-bound RT cutoff. We also correlated both hit rates and stimulus-driven behavioral modulation scores across participants with and without applying a lower-bound RT cutoff. We found that hit rates were correlated $r = .9954$ and stimulus-driven behavioral modulation scores were correlated $r = .9595$. We have opted not to reanalyze the neural data including this cutoff because of the small number of excluded trials and strong correlations between dependent measures with and without use of the cutoff. We hope that the Reviewer supports this decision.

Figure R1. Behavioral data with and without use of a lower-bound (LB) reaction time cutoff. Left: Histogram shows the number of trials excluded with use of a lower-bound cutoff (x-axis). All but 2 participants had a discrepancy of 5 trials or less (2% of total trials). Middle: Hit rates were strongly correlated ($r = .9954$) with and without use of a lower-bound cutoff, as were stimulus-driven behavioral modulation values (Right).

The authors may want to consider citing one of these recent papers as additional evidence of age-related cortical over-representation in longer duration speech samples (p. 12), perhaps in the context the implications of the current work for naturalistic speech processing:

Presacco, A., Simon, J. Z., & Anderson, S. (in press). Evidence of degraded representation of speech in noise, in the aging midbrain and cortex. *Journal of Neurophysiology*. doi:10.1152/jn.00372.2016

Presacco, A., Simon, J. Z., & Anderson, S. (in press). Effect of informational content of noise on speech representation in the aging midbrain and cortex. *Journal of Neurophysiology*. doi:10.1152/jn.00373.2016

Thanks for pointing these papers out to us. We now cite them in our discussion of the comparison between age effects on entrainment (for which older adults' responses were reduced) versus age effects on ERPs to tone onsets (for which older adults' N1s were enhanced). We have also included a brief discussion on limitations of our study (in response to the Reviewer's earlier comment), in which we discuss the benefits of using naturalistic speech (and again cite these two papers). The relevant revisions are reproduced below for the Reviewer's convenience.

pp. 15–16, Discussion, Entrainment to frequency modulation becomes weaker and less flexible with age: "Moreover, recent work using an approach that involved reconstructing speech envelopes from brain responses showed exaggerated cortical representations of speech compared to younger adults, despite depressed brainstem responses^{65,66}; for older adults, speech-envelope reconstruction accuracy was negatively correlated with Flanker task scores⁶⁵, reinforcing the idea that older adults' aberrant cortical response magnitudes may stem from an excitation–inhibition imbalance⁶⁷."

p. 18, Discussion, Future directions and limitations: "We started out with the notion that age-related changes in either measure might contribute to listening and speech comprehension difficulties, and in particular difficulty solving the cocktail-party problem. However, the current study neither makes use of naturalistic speech stimuli^{65,66}, nor does it attempt to recreate the cocktail-party problem by including distractor sounds."

Reviewer #4 (Remarks to the Author):

Henry and colleagues in their paper „The balance between rhythmic neural entrainment and top-down neural modulation changes with age“ study the ability of healthy younger and older subjects to detect short gaps in sound trains. For this purpose, they use a paradigm very similar to their previous study (Henry and Obleser2012), in which they demonstrated that the successful detection of the gap depends on the delta phase at the CZ electrode. In the present paper, they now compare differences in entrainment frequency and alpha amplitude between young and older subjects. The major finding is that the older subjects suppress the alpha oscillations more than the younger subjects, while the entrainment is less pronounced in the older subjects. The comparison of older and younger subjects is intriguing in trying to explain why older subjects have more difficulties in separating multiple sounds. These results are of interest for researchers studying the auditory domain and might also have interesting implication for studies on ageing.

While the paper is well written, the methods description is rather on the short side, leaving many technical questions open that are crucial for judging the validity of the results. For that reason, some of the results are not convincing without further analysis and explanation. For example, clear information from which EEG electrode the different results are coming from at each step is missing and makes the paper rather confusing to read.

Additionally, the paper and its claims would have greatly benefited from using high-density EEG or MEG and then performing the analysis on the cortical surface. Such an analysis would allow identifying neural generators, which the authors discuss, but cannot pinpoint down. I understand that this would require a complete new experiment and is therefore most likely beyond the scope of the present paper. The low resolution and the analysis in the sensor space nevertheless negatively impact the value added for the research community.

We thank the reviewer for the positive remarks regarding the writing and implications of our manuscript. We have happily tried to clarify and expand on methods descriptions in the hopes that the validity of our results is now clearer.

With respect to the use of high-density EEG or MEG, it is important to note that we are not making claims regarding the precise locations of cortical generators of any of the examined neural responses. Rather we are simply providing evidence that generators for entrained responses versus ERPs are dissociable, a claim which has been made stronger based on analysis tweaks suggested by Reviewer 2. We think (and hope the Reviewer will agree with us on this) that the most valuable contribution of the manuscript – the demonstration of a trade-off between sensory-driven entrained responses and top-down neural modulation – does not rest on the spatial resolution of our recording technique. We hope to have clarified the methods, and thus the contribution of our manuscript, for the Reviewer satisfactorily.

I will detail my concerns in the following:

Major Points:

1) Figure 3 and corresponding methods: Which baseline was used? The baseline – if it is the one plotted in the ERP plots – is much lower for the younger subjects. This could largely explain the effect of the amplitude differences in young and older subjects. Therefore at the moment I cannot judge whether the described ERP effect is a true one or related to an inappropriate baseline choice.

Good question – we had completely omitted this information from the Methods section. We actually did not baseline correct the ERPs; we instead applied a high-pass filter (0.6 Hz, 1395 points, Hann window) to remove slow drifts. We hope that this satisfies the reviewer that our baseline choice could not have caused the age-related differences in ERP component amplitudes, since we did not apply a correction.

To be sure that the details of the ERP analysis did not cause any of the observed age differences, we also reanalyzed the ERP data after baseline correcting based on the -0.05 s to 0 s (relative to tone onset) time window. Using this baseline correction did statistically strengthen the age difference for the N1 component ($t(38) = 4.47, p = .00007$), in line with the reviewer's intuition. It slightly weakened the effect on the P2 component, which became marginally significant ($t(38) = 1.83, p = .07$), and did not change the null P1 effect ($t(38) = 0.30, p = .77$).

We have now added the following text and references to the manuscript:

pp. 28–29, Methods, Data acquisition and analysis, ERP and topography analysis: “ERPs were not baseline corrected since they were high-pass filtered to remove slow drifts (however, all results were consistent with those reported in the Results section when a baseline ranging from -.05 s to 0 s prior to tone onset was subtracted from the ERPs). High-pass filtering instead of baseline correction is particularly well suited for fast presentation designs as employed in the present study⁸⁸⁻⁹¹.”

Some further labels on the ERP plots would be very helpful as well as on the small bar graph inlays. Which electrodes were included in the ERP analysis?

We have added labels on the ERPs plots in Figure 3. The new figure is reproduced on p. 6 of this letter, along with the figure caption, which has been updated to include information about which electrodes were used in the analysis (Fz, F3, F4, FC3, FC4, Cz, C3, C4).

We have also added electrode information to the Methods section.

p. 27, Methods, Data acquisition and analysis, ERP and topography analysis: “For analysis of ERPs, time-domain data from a fronto-central electrode cluster (Fz, F3, F4, FC3, FC4, Cz, C3, C4) were low-pass filtered at 20 Hz (6th-order Butterworth) and averaged.”

2) For the phase-estimation it is essential to see that the results are unaffected by the Wavelet filtering (Zoefel and Heil 2013). Even though the ERP is removed, I still would be more convinced that there is a true phase effect if the analysis would be also done with an acausal filter.

We recently conducted a simulation analysis (in response to a very similar Reviewer comment), the results of which are published in the Methods section of Henry, Herrmann, and Obleser (2016). For this analysis, we simulated single-trial EEG signals, for which we knew the ground-truth phase value in the time window just before gap onset (similar to the current study, we averaged phase values in a frequency-dependent time window corresponding to 10% of a cycle). Similar to the procedure of Zoefel and Heil (2013), we then added the grand-average hit ERP or the grand-average miss ERP to each single-trial's post-gap interval (which of course exaggerates any problems of phase distortion, as single-trial ERPs are more variable than the grand average). We then applied our windowing technique to each single trial and again estimated the phase in a frequency-dependent time window corresponding to 10% of a cycle. We compared the pre- and post-ERP-cleaned phase values on a single-trial basis.

We found that this technique introduced a bias of -0.22 radians that was identical in magnitude for hit and miss trials, and that did not depend on frequency within the tested range of 1–15 Hz (p. 862, Henry et al., 2016). Thus, the method does not introduce a systematic phase difference between detected and undetected gaps that could have influenced the current results.

3) Regarding the result of no correlation between hit rate and neural amplitude for both subject groups together: How can this be explained in relation to the previous results in young subjects (Henry and Obleser 2012), where a correlation was present. I understand that when controlling for alpha amplitude a partial correlation is found, but that was not necessary in the previous study. Therefore, the results seem to contradict each other.

We have actually never calculated a correlation between hit rate and neural amplitude in any of our previous studies. We have used circular–linear correlations to demonstrate a relationship between stimulus/neural phase and hit rates. In Henry and Obleser (2012), we also used a circular–circular correlation to demonstrate that individual differences in phase lag between stimulus phase and hit rates could be explained by knowing both the phase lag between neural phase and stimulus as well as the phase lag between stimulus phase and neural phase. The current manuscript is the first time we have attempted to relate the strength of entrainment (neural amplitude) to the degree to which behavior depended on neural phase.

4) On page 8 2 subjects are excluded from analysis. From which group were the 2 subjects excluded? Was the 1 subject that was excluded from the later analysis among them? Were the 2 subjects included in the following analysis?

Both excluded participants were from the older group. However, in the revised manuscript we use nonparametric statistics for that analysis (of topographies), and have included all participants. The single participant that was excluded from the correlation analyses was from the young group. We have now added these details to the manuscript.

Minor Points:

5) Page 5: Stimulus-driven behavioral modulation: Is this the measure described on page 19? Here it is not clear which particular kind of smoothing kernel is taken and how this choice is justified. At the same time, these are only behavioral data for which the minimum and maximum are taken so that smoothing seems unnecessary. In Figure 1c the y-axis labels are missing. Particularly for the last plot it is not obvious what is plotted there.

Yes, this is the measure described on p. 19 (now p. 23 in the revision). The data were smoothed with a 5-bin unweighted kernel.

p. 24, Methods, Data acquisition and analysis, Behavioral data analysis: “The degree to which hit rates were modulated by FM phase (referred to as “stimulus-driven behavioral modulation”) was calculated as the maximum minus the minimum hit rate over the 20 FM phase bins after data were smoothed with a 5-bin unweighted kernel.”

It is important to smooth these data because a participant could have a very large maximum–minimum value due simply to noisy data, however we were interested in systematic relationships between stimulus/neural phase and hit rate. Smoothing penalizes extreme values that are unrelated to their neighbors more than those that continue a trend from their neighbors. We have added this explanation to the manuscript.

p. 24, Methods, Data acquisition and analysis, Behavioral data analysis: “It is important to smooth these data because an individual could have a very large maximum–minimum value due simply to noisy data, however we were interested in systematic relationships between stimulus phase and hit rate. Smoothing penalizes extreme values that are unrelated to neighboring data points more than those that continue a trend from their neighbors.”

We note that hit rates as a function of neural phase were also smoothed, but this was accomplished by sorting single trials into overlapping bins based on neural phase. The reason that the strategies differ for the two phase measures (i.e., stimulus phase and neural phase; see also comment #13) is that gaps were placed into 20 discrete locations around the stimulus cycle, whereas neural phase is a continuous measure.

With respect to Figure 1C, we have now added axis labels (which had previously been titles, but we agree that was confusing). The new figure is reproduced here for the Reviewer’s convenience:

Figure 1. Entrained neural responses of younger and older listeners were tested using FM stimuli and tone sequences. **A)** Stimuli and experimental design. Participants were exposed to two blocks of FM stimulation; during an *active* block, they detected near-threshold gaps, while during a *passive* block they ignored the same stimuli (in counterbalanced order). Between FM blocks, participants passively listened to an 8-minute block of tone stimulation (1400 tones), with presentation rate and frequency range matched to the rate and modulation depth of the FM stimuli. **B)** Audiograms for younger (blue) and older (purple) listeners shown separately for the left (top) and right (bottom) ears. Both participant groups had average hearing thresholds better than 20 dB up to 2000 Hz. **C)** Younger and older listeners did not differ significantly according to any behavioral measure.

6) Figure 2B: Please specify for which frequency the amplitudes are taken (2.8 Hz?). I guess that the amplitude is calculated from the fronto-central electrodes described on page 21.

Yes to both. We have now included this information in the Figure 2 caption:

p. 8, Figure 2 caption: “Figure 2. Attention effects on neural entrainment strength were larger for younger than for older adults. A) Frequency-domain representations of the EEG signal for active (pink) and passive (blue) listening to FM stimuli, shown separately for younger (left) and older (adults) averaged over a fronto-central electrode cluster: Fz, F3, F4, FC3, FC4, Cz, C3, C4. **B)** 2.8-Hz spectral amplitude: A significant interaction between Age and Attention resulted from stronger effects of attention (active minus passive) for younger as compared to older adults ($p = .04$).”

7) Figure 4: A y-axis would be helpful.

We have added a y-axis. The figure is reproduced on p. 8 of this letter.

8) Page 9: Which Figure is meant by 3B? There is no figure 3B.

Thanks for that. A leftover from a previous version in which there was a Figure 3B.

9) Figure 5 -7: The measure “hit rate (peak-trough)” is not explained. Is it the stimulus-driven

behavior modulation described on page 19? Also the residual of this cannot be understood without further explanation. Units on the y-axis are missing.

Figures 5–7 show neuro-driven behavioral modulation rather than stimulus-driven behavioral modulation. We have updated the figures and captions so that this is clear. We reproduce the figures and captions below for the Reviewer’s convenience. We have also added a more thorough description of the residuals in the Figure 7 caption. We note that both stimulus-driven and neuro-driven behavioral modulation are unitless measures, as they are ratios and the units cancel during division.

Figure 5:

Figure 5. Gap detection is modulated by pre-gap neural phase. Left) Neuro-driven behavioral modulation: Hit rates (shown zero-centered) were modulated by 2.8-Hz neural phase with a similar magnitude (neuro-driven behavioral modulation, i.e., hit rate peak-trough) for younger and older participants (inset). **Right)** Neuro-driven behavioral modulation was not correlated with entrainment strength. Best-fit line shown ignoring outlying data point (from the younger group). Inset plots ranks on which Spearman correlation was based. Rank on 2.8-Hz amplitude is plotted on x-axis and rank on neuro-driven behavioral modulation is plotted on y-axis.

Figure 6:

Figure 6. Alpha amplitude dynamics differed between younger and older adults. Left) Younger adults showed a significant increase in alpha amplitude during active task performance compared to passive listening (left), while older adults (**Middle**) showed the opposite effect. FDR-corrected significance is shown for the main effects of attention (black) and age \times attention interaction (dark gray) on plots of both the younger and older alpha time courses. Simple effects of attention within each age group (light gray) are shown separately for younger and older data. **Right)** Alpha amplitude averaged over the 10-s stimulus time course from electrode Pz correlated positively with 2.8-Hz amplitude averaged over a fronto-central electrode cluster (Fz, F3, F4, FC3, FC4, Cz, C3, C4; top) and negatively with neuro-driven behavioral modulation (bottom). Insets show ranks on which Spearman correlations were based. Ranks on alpha amplitude are plotted on the x-axis. Y-axis shows ranks on 2.8-Hz amplitude (top) and neuro-driven behavioral modulation (bottom).

Figure 7:

Figure 7. Entrainment strength predicts neuro-driven behavioral modulation if alpha amplitude is taken into account. A) The partial correlation after partialling out alpha amplitude from the Pz electrode between 2.8-Hz amplitude (averaged over a fronto-central electrode cluster: Fz, F3, F4, FC3, FC4, Cz, C3, C4) and neuro-driven behavioral modulation was significant. Plot shows residuals i.e., deviations of the neuro-driven behavioral modulation observations from the fitted function characterizing the relationship between alpha amplitude and neuro-driven behavioral modulation, plotted against the residuals from the correlation between alpha amplitude and 2.8-Hz amplitude. Best-fit lines are shown separately for younger and older groups. Line for younger group excludes the extreme data point marked with arrow. **B)** Schematic illustrating the statistical interaction between neural entrainment, alpha amplitude, and behavioral modulation. Behavioral modulation was significantly predicted by neural entrainment strength, but only when alpha amplitude was partialled out. Inset shows ranks on which Spearman correlation was based. X-axis shows ranks on 2.8-Hz amplitude residuals, and y-axis shows ranks on neuro-driven behavioral modulation residuals.

10) Figure 7: Across which electrodes was the alpha amplitude taken? Is it Pz as mentioned in the methods section?

Yes. We have added this missing information to the Figure 7 caption (reproduced in response to comment #9).

11) Please also provide the mean age and standard deviation besides the range.

Done.

p. 22, Methods, Participants: “Forty individuals (20 younger [10 male, 10 female], age 18–31 years, $M = 25.4$ years \pm $SD = 3.3$ years; 20 older [8 male, 12 female], age 61–77 years, $M = 67.3$ years \pm $SD = 5.3$ years) took part in the experiment.”

12) Page 17: SL has not been introduced.

Fixed.

p. 22, Methods, Stimuli: “All stimuli were normalized with respect to peak amplitude, and were presented at 50 dB sensation level (SL; hearing thresholds were determined prior to EEG recordings).”

13) Page 22: Why are 14 phase bins chosen for the neuro-driven behavioral modulation while 20 bins are chosen for the stimulus driven behavioral modulation? For comparison the same number of bins would be more appropriate.

The reason that the strategies differ for the two phase measures (i.e., stimulus phase and neural phase) is that gaps were placed into 20 discrete locations around the stimulus cycle, whereas neural phase is a continuous measure. This means that gap locations and (lack of overlap) were fixed for stimulus phase based on the design of the stimuli. However, when we analyze neural phase, we prefer to overlap bins as a method of smoothing.

The question raised by the Reviewer is a good one though: how much does the choice to bin differently affect the neural phase analysis? In order to estimate the impact of our analysis choices, we calculated neuro-driven behavioral modulation in two ways – first, as we report in the manuscript based on 14 overlapping phase bins; and second, based on 20 nonoverlapping phase bins (data were then smoothed so that data were treated identically to the calculation of stimulus-driven behavioral modulation). First, we correlated the two resulting measures of neuro-driven behavioral modulation. The correlation was very strong and significant ($r = .79$, $p < .00001$), indicating that the two pipelines lead to similar estimates of neuro-driven behavioral modulation (see Fig. R2). Second, we calculated a difference score by subtracting the estimates of neuro-driven behavioral modulation obtained using the two different binning strategies. Difference scores did not differ from zero for either age group (young: $t(19) = 1.10$, $p = .29$; old: $t(19) = -0.59$, $p = .56$), and did not differ as a function of age group ($t(38) = 1.24$, $p = .22$). Thus, estimates of neuro-driven behavioral modulation were not systematically biased based on our analysis decisions for either group (Fig. R2).

We hope, based on these results, to have the Reviewer’s support in reporting the phase analyses as they appeared in the original manuscript, rather than replicating every analysis in the paper with a measure that is highly correlated with the one we initially reported.

Figure R2. Left) Estimates of neuro-driven behavioral modulation calculated two ways (one using 14 overlapping bins and the other using 20 nonoverlapping bins) were highly correlated ($r = .79$, $p < .00001$). **Right)** Difference scores calculated by subtracting the estimates of neuro-driven behavioral modulation (14 bins – 20 bins) were not different from zero for either group ($p \geq .29$), and were not different between age groups ($p = .22$).

References:

- Henry, M. J., Herrmann, B., & Obleser, J. (2016). Neural microstates govern perception of auditory input without rhythmic structure. *The Journal of Neuroscience*, *36*, 860-871.
- Henry, M. J., & Obleser, J. (2012). Frequency modulation entrains slow neural oscillations and optimizes human listening behavior. *Proceedings of the National Academy of Sciences USA*, *109*, 20095-20100.
- Rihs, T. A., Michel, C. M., & Thut, G. (2009). A bias for posterior α -band power suppression versus enhancement during shifting versus maintenance of spatial attention. *NeuroImage*, *44*, 190-199.
- Zoefel, B., & Heil, P. (2013). Detection of near-threshold sounds is independent of EEG phase in common frequency bands. *Frontiers in Psychology*, *4*.

REVIEWERS' COMMENTS:

Reviewer #1 (Remarks to the Author):

The authors have reframed and revised sections of the manuscript, resulting in improved clarity. I appreciated the additional information regarding correlations with age groups, but I agree that the inclusion of these results would detract from the manuscript.

As I mentioned before, this manuscript describes an interesting approach to assessing sensory and cognitive changes with aging and significantly advances our understanding of the mechanisms underlying decreased speech understanding in noise in older adults.

Reviewer #2 (Remarks to the Author):

The authors have included all my comments in their substantial revisions. I think this has improved the paper which is also well written/ a pleasure to read. A very interesting case (convincing in my view) is made that for upholding auditory/ sensory sampling mechanisms in a physiological useful range, young and elderly participant use different strategies reflected in different oscillatory dynamics and in differential weights given to continuous (inhibitory) and rhythmic (entrainment) processes associates with sensory sampling. This adds significantly to the literature (refines current knowledge), also because starting to reveal individual differences (here by age).

I have no further suggestions for improvements.

Reviewer #3 (Remarks to the Author):

I thank the authors for their careful and thorough review of my comments. They have successfully addressed my concerns.

Reviewer #4 (Remarks to the Author):

The authors have answered all my questions.

I have two points, which should be clarified:

1) At what point were the data averaged for the 2.8 Hz amplitude across the fronto-central electrode cluster?

2) On page 26 it is stated that shorter epochs from -1.95 to 1.95s were extracted. But before the epoch length was -1.5 to 1.5s for the second pipeline and -1.6/-1.5 to 1.9/1.5s. Please clarify, because the shorter epochs seem to be longer either at the baseline, the post-stimulus interval or both.

Response to Reviewer

Below, we provide point-by-point responses to the remaining concerns of Reviewer 4.

**The authors have answered all my questions.
I have two points, which should be clarified:**

1) At what point were the data averaged for the 2.8 Hz amplitude across the fronto-central electrode cluster?

The data were averaged over electrodes *after* performing the FFT on trial-averaged time-domain data at each electrode separately. We have clarified this in the Methods:

p. 22, Methods, EEG data acquisition and analysis: “Subsequently, frequency-domain data were averaged over a fronto-central electrode cluster comprising electrodes Fz, F3, F4, FC3, FC4, Cz, C3, and C4.”

2) On page 26 it is stated that shorter epochs from -1.95 to 1.95s were extracted. But before the epoch length was -1.5 to 1.5s for the second pipeline and -1.6/-1.5 to 1.9/11.5s. Please clarify, because the shorter epochs seem to be longer either at the baseline, the post-stimulus interval or both.

Epochs for different analyses were indeed different lengths, but it is important that each time data were epoched, we were starting from the raw (or filtered) *continuous* data. We have clarified this in the Methods:

p. 21, Methods, EEG data acquisition and analysis: “The first pipeline involved high-pass filtering the continuous raw data from each block (active FM, passive FM, passive tone sequence; 0.6 Hz, 1395 points, Hann window). Data were then divided into individual trial epochs, which for the FM blocks ranged from -1.5 to 11.5 s with respect to FM-stimulus onset and for the tone-sequence block ranged from -1.6 to 1.9 s with respect to each tone onset.”

p. 21, Methods, EEG data acquisition and analysis: “The second pipeline, geared towards the analysis of pre-stimulus 2.8-Hz phase, critically omitted high-pass filtering, and thus first involved epoching the continuous raw data from the active and passive FM blocks (-1.5 to 15.5 s) and then low-pass filtering (51 points, Hann window), re-referencing, and removal of the same components as the previously described pipeline using ICA.”